# Sigma-1 receptor chaperones rescue nucleocytoplasmic transport deficit seen in cellular and *Drosophila* ALS/FTD models

Pin-Tse Lee [1,2,5,6], Jean-Charles Liévens [3,6], Shao-Ming Wang [1,6], Jian-Ying Chuang [2], Bilal Khalil [4], Hsiang-en Wu [1], Wen-Chang Chang [2], Tangui Maurice [3] & Tsung-Ping Su [1✉]

In a subgroup of patients with amyotrophic lateral sclerosis (ALS)/Frontotemporal dementia (FTD), the (G4C2)-RNA repeat expansion from C9orf72 chromosome binds to the Ran-activating protein (RanGAP) at the nuclear pore, resulting in nucleocytoplasmic transport deficit and accumulation of Ran in the cytosol. Here, we found that the sigma-1 receptor (Sig-1R), a molecular chaperone, reverses the pathological effects of (G4C2)-RNA repeats in cell lines and in *Drosophila*. The Sig-1R colocalizes with RanGAP and nuclear pore proteins (Nups) and stabilizes the latter. Interestingly, Sig-1Rs directly bind (G4C2)-RNA repeats. Overexpression of Sig-1Rs rescues, whereas the Sig-1R knockout exacerbates, the (G4C2)-RNA repeats-induced aberrant cytoplasmic accumulation of Ran. In *Drosophila*, Sig-1R (but not the Sig-1R-E102Q mutant) overexpression reverses eye necrosis, climbing deficit, and firing discharge caused by (G4C2)-RNA repeats. These results on a molecular chaperone at the nuclear pore suggest that Sig-1Rs may benefit patients with C9orf72 ALS/FTD by chaperoning the nuclear pore assembly and sponging away deleterious (G4C2)-RNA repeats.

[1] Cellular Pathobiology Section, Integrative Neuroscience Research Branch, Intramural Research Program, National Institute on Drug Abuse, NIH, 333 Cassell Drive, Baltimore, MD 21224, USA. [2] The Ph.D Program for Neural Regenerative Medicine, Taipei Medical University, 250 Wuxing Street, Taipei 11031, Taiwan. [3] MMDN, University of Montpellier, EPHE, INSERM, Montpellier, France. [4] Department of Neuroscience, Mayo Clinic, Jacksonville, FL 32224, USA. [5] Present address: Taiwan Biomaterial Company, 6F, No. 26-1, Sec. 2, Shengyi Rd., Zhubei City, Hsin-Chu County 30261, Taiwan. [6] Those authors contributed equally: Pin-Tse Lee, Jean-Charles Liévens, Shao-Ming Wang. ✉email: TSU@intra.nida.nih.gov

Amyotrophic lateral sclerosis (ALS) or frontotemporal dementia (FTD), either sporadic or familial, is a devastating neurological disease that currently has no cure. The exact molecular mechanisms which lead to this disease remain to be fully clarified. Two studies in 2011 discovered that the (G4C2)-RNA hexanucleotide repeat expansion (HRE) upstream of the start codon of the C9orf72 gene plays a critical role in the familial ALS[1,2] and FTD[3]. The (G4C2)-RNA repeats in normal subjects range between 3 and 20. In ALS/FTD patients those repeats can be up to hundred or thousand. It is known that HRE can form G-quadruplex structures through the intermolecular hydrogen bonding between guanines[4]. Exactly how the HRE causes ALS/FTD is a very active area of research[5–10].

A study demonstrated that the HRE causes the nucleolar stress resulting in the diffusion of an essential component of nucleoli, nucleolin, to disperse throughout the nucleus[11]. This result suggests an interaction between nucleolin and HRE in C9orf72 ALS patients and suggests the nucleolar stress as an underlying mechanism of this disease. Another study[12] indicated that the HRE binds to RanGAP and in doing so impedes the activation of RanGTP (i.e., Ran-GTPase (referred to as Ran in this report) in the form of GTP). Ran is a small Ras-related GTPase that mediates the nucleocytoplasmic exchange of macromolecules across the nuclear envelope. Normally, RanGTP needs to be activated by RanGTP-activating protein (RanGAP) at the cytosolic side of the nuclear pore before it can be hydrolyzed to RanGDP to firstly provide energy for an effective transport of cargos from cytosol into nucleus and secondly to allow itself to be transported into nucleus. Once inside the nucleus, RanGDP is converted by guanine exchange factor into RanGTP which is then transported back to cytosol so that it can be activated by RanGAP to initiate a new round of cycle to facilitate cargo entries into the nucleus[12,13]. In C9orf72 patients, as a result of this action of HRE on RanGAP, Ran is heavily accumulated in the cytoplasm, reflecting a pathological nucleus/cytoplasmic gradient of Ran as well as a deficient nucleocytoplasmic transport in ALS[12,13]. The nuclear pore that Ran passes through between cytosol and nucleus is called nuclear pore complex (NPC) that is a megadalton structure[14,15] at the nuclear membrane. The proteins that make up the NPC are called nucleoporins (Nups) that is composed of at least 34 distinct constituent proteins[14,15]. Some of Nups are facing cytosol, at the midportion of pore, or facing the nucleus internal[14,15]. The stability of Nups may relate to the integrity of the nucleocytoplasmic transport. It has to be mentioned that the nucleocytoplasmic transport deficit as seen in the maldistribution of Ran is also an important factor in frontotemporal dementia (FTD)[3]. Thus understanding the fundamental mechanisms controlling the transport is critical not only to ALS but also to FTD.

The Sig-1R[16–25] is a ligand-regulated molecular chaperone that resides mainly at the mitochondria-endoplasmic reticulum (ER) interface, referred to as the mitochondria-associated ER membrane (MAM), where it chaperones IP3R3 to ensure proper $Ca^{2+}$ signaling from the endoplasmic reticulum into mitochondria[26,27]. The Sig-1R exists at other parts of cell as well and has been proposed to be an important protein for cellular survival[28–34] as a dynamic pluripotent modulator in living systems[35].

Different type of cells has been shown to exhibit different subcellular localization of Sig-1Rs. For example, electron-microscopic studies show that while the Sig-1R exists on the plasma membrane in the dorsal root ganglia[36], it exists however only inside of retinal neurons[37]. Interestingly, the Sig-1R exists at the nucleoplasmic reticulum[38] as well as at the nuclear envelope of neurons[39] where the Sig-1R binds emerin to recruit chromatin-remodeling molecules and regulates gene transcription[39].

The Sig-1R has been reported to relate to the familial ALS[40,41]. In animal model, a study suggested that a loss of function of Sig-

1Rs at the MAM might lead to dysfunctional ER-mitochondrion crosstalk and thus the ALS[42]. Expression of Sig-1R with E102Q mutation recapitulates ALS pathology in Drosophila[43]. A lack of Sig-1R has been shown to exacerbate ALS progression in a mouse model of ALS[44,45]. A drug targeting Sig-1R has been shown to be effective in an animal model of ALS[30]. Other potential mechanisms, if any, underlying the role of Sig-1Rs in ALS remain to be fully established.

Here we found that in cellular models the Sig-1R exists at the NPC where it counteracts the aberrant nucleocytoplasmic distribution of Ran, caused by the (G4C2)-RNA repeats, by chaperoning Nups and by sponging away the toxic (G4C2)-RNA repeats. Further, we extend the functional readouts of the biochemical and cellular biological data obtained in cell lines to a Drosophila model and validate that the Sig-1R but not its E102Q mutant reverses morphological, behavioral, and electrophysiological deficits caused by the (G4C2)-RNA repeats. Those results are presented in this report.

## Results

**The Sig-1R exists at the nuclear pore**. Our previous report demonstrated the existence of Sig-1Rs at the nuclear envelope in proximity to RanBP2[39]. We examined here if the Sig-1R may exist at the nuclear pore in HeLa cells. Immunocytohistochemistry indicates the colocalization of HA-tagged Sig-1Rs with endogenous RanGAP and nuclear pore proteins Nup62 and Nup358 (i.e., RanBP2) (Fig. 1a). In immunoprecipitation (IP) assay, GFP-tagged-Sig-1Rs co-IP with RanGAP and Ran (Fig. 1b). With mAb414 as the blotting antibody, which is well-known to recognize FG-repeat-Nups, Nups co-IP with V5-tagged Sig-1R (Fig. 1c). Anti-Nup50 antibody also co-IPs the endogenous Sig-1R (Fig. 1d). Note: similar to that seen with HA-Sig-1R, the endogenous Sig-1R colocalizes with RanGAP1, Nup62, and NuP358 (Supplementary Fig. S1).

NSC-34 motor neuron-like cells (spinal × neuroblastoma hybrid cells) are often used as a bona fide cellular model to investigate the physiopathological mechanisms of ALS[46,47]. Indeed, HA-Sig-1Rs colocalize with RanGAP and Nup62 at the nuclear membrane in NSC-34 cells (Fig. 2). HA-tagged Sig-1Rs colocalize with endogenous RanGAP and Nup62 at the nuclear membrane in NSC-34 cells (Fig. 2a). Also, three-dimensional (3D) images acquired by sequentially capturing a series of 2D sections were performed to further confirm the nuclear envelope localization of HA-Sig-1Rs in NSC-34 cells (Fig. 2b, c). Results indicated that HA-Sig-1Rs colocalize with RanGAP in sections 3–20 (Fig. 2b) and Nup62 in sections 3–18 (Fig. 2c), specifically indicated as such at section 11 in the cross-sectional intensity scanning (the central panels of Fig. 2b, c).

Those results suggest that the Sig-1R exists at the nuclear pore where it interacts with RanGAP and certain Nups.

**Sig-1R stabilizes the nuclear pore proteins**. Inasmuch as the Sig-1R is a molecular chaperone in chaperoning IP3 receptor[26] as well as IRE-1[48], we examined if the Sig-1R may influence the stability of de novo Nups in HeLa cells. Indeed, cells treated with shSig-1R for 48 h to knockdown Sig-1Rs show a reduction of Nup358, Nup214, Nup62 (Fig. 3a) and Nup50 (Fig. 3b). Turnover of Nups was then examined in a time-lapsed manner in cycloheximide-treated cells in which cycloheximide was used to stop the de novo synthesis of proteins. Cycloheximide is known to interfere with the translation step in protein synthesis, thus blocking translational elongation. In the presence of cycloheximide, all proteins detected by western blots are existing proteins waiting to be degraded without the presence of newly de novo synthesized proteins. Thus, western blots

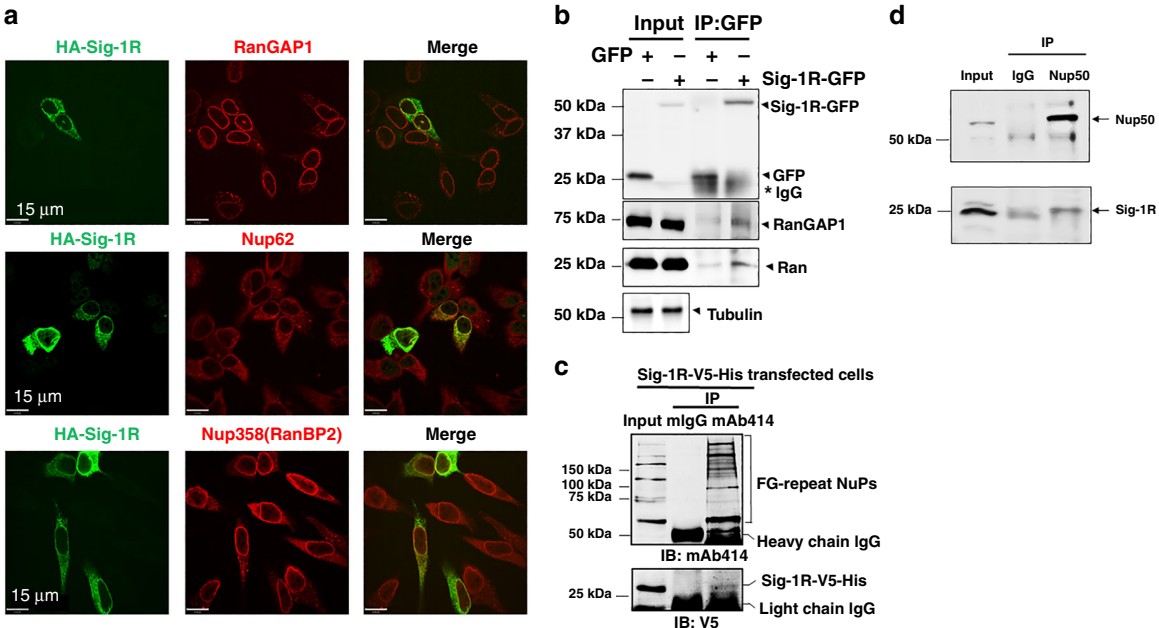

**Fig. 1 Localization and association of HA-tagged Sig-1R, RanGAP1, and a nuclear pore complex proteins Nup62 in HeLa cells. a** Immunohistochemistry followed by confocal microscopic examination indicates perinuclear colocalizations of immunoreactive HA-tagged Sig-1R (green) with RanGAP (red), Nup62 (red) and Nup358 (red). HeLa cells transiently transfected with human HA-Sig-1R vectors were used. **b** Coimmunoprecipitation (Co-IP) of GFP-tagged Sig-1R with RanGAP and Ran. HeLa cells were transfected with GFP or GFP-Sig-1R vectors for 24 h before the co-IP experiment. Proteins interacting with GFP (control) or GFP-Sig-1R were detected by western blot. **c** NuPs' interaction with Sig-1R in a co-IP experiment. The mAb414 pulled down FG repeats-containing Nups together with Sig-1R-V5-His which was transfected into HeLa cells. **d** Nup50 antibody co-IPed with endogenous Sig-1R which in this experiment was detected by the Santa Cruz B5 anti-Sig-1R antibody (sc137075). Note: all other endogenous Sig-1Rs in western blot in the cell line portion of this study was detected by custom-made anti-Sig-1R antiserum #5460 (see Methods section). The two Sig-1R antibodies have been used interchangeably in the lab to reserve #5460 which is custom-made polyclonal and is limited in quantity (see Methods section). Note: Santa Cruz B5 anti-Sig-1R is monoclonal, thus almost unlimited. Note: colocalization of endogenous Sig-1R with RanGAP1 and Nup62 in HeLa cells is shown in Supplementary Fig. S1. Sig-1R, Sigma-1 receptor, RanGAP RanGTP-activating protein, Nup nucleoporin. $n = 4$ (**a**), $n = 4$ (**b**), $n = 3$ (**c**), and $n = 3$ (**d**) independent experiments with similar results each from biologically independent cells or cellular preparations.

typically show a time-dependent decrease of that protein of interest. Western blotting indeed indicates a time-dependent decrease of Nup358 and Nup214 between 4 and 8 h after cycloheximide (100 µg/ml) treatment (Fig. 3c). A representative western blot result from the Nup50 turnover is shown in supplemental information (Supplemental Information, Fig. S2). Note that 150 µg/ml of cycloheximide was needed to see the decrease of de novo synthesized Nup50 (Supplemental Information, Fig. S2), suggesting a relatively stable nature of Nup50 when compared to other Nups. Summarized results from multiple independent time-lapsed experiments indicate that the Sig-1R knockdown significantly decreases the stability of Nup358, Nup214, and Nup50 (Fig. 3d–f). PCR (Supplemental Information, Fig. S3a) and Real-time PCR (Supplemental Information, Fig. S3b) indicate that mRNA levels of those Nups are not affected by the knockdown of Sig-1Rs. Those results suggest that the Sig-1R chaperone stabilizes Nups at the post-translational level.

**Purified Sig-1R binds the (G4C2)-RNA repeats**. The Sig-1R has been shown to bind proteins[23,35] and lipids[49,50], suggesting that the Sig-1R may accommodate diverse chemical natures of its binding partners. Thus, it is not unreasonable to speculate that the Sig-1R might bind RNA as well.

Firstly, we used immunostaining to examine if Sig-1Rs might colocalize with $(G4C2)_{31}$-RNA[5] that were transfected into HeLa cells. The RNA fluorescence in situ hybridization (RNA FISH)

technique was used to detect the $(G4C2)_{31}$-RNA[5]. Indeed, the HA-Sig-1R detected by the HA antibody colocalizes with $(G4C2)_{31}$-RNA, especially in the perinuclear area (Fig. S4a). The specificity of the RNA signal in FISH was verified by the DNase and RNase in that only the RNase abolished the FISH signal (Fig. S4b).

Next, we examined if purified Sig-1Rs might bind biotinylated $(G4C2)_{10}$-RNA in an assay carried out in test tube containing purified molecules. The assay conditions are shown in supplementary data (Supplemental Information, Fig. S5a). Purified human GST-Sig-1R and GST are also shown (Supplemental Information, Fig. S5b, c). Results indicated that purified GST-human Sig-1R (lane 4, Fig. 4a) but not GST (lane 3, Fig. 4a) directly binds Biotin-$(G4C2)_{10}$-RNA. Much clearer result was obtained when rat microsomes, which are highly enriched in Sig-1Rs, were solubilized and incubated with Biotin-$(G4C2)_{10}$-RNA for the assay (Fig. 4b). This is because of the mass action law governing that the increased concentration of Sig-1Rs in liver microsomal preparation will bind more to the same concentration of Biotin-$(G4C2)_{10}$-RNA. When scrambled RNA (i.e., Biotin-$(A2U2GC)_{10}$-RNA) was used as control, it failed to bind to the mouse Sig-1R-YFP (Fig. 4c, lane 8, note: compared to lane 4). Single amino acid mutation of Sig-1R at amino acid 102 from glutamic acid to glutamine was reported to relate to familial ALS[41]. We examined if this mutant of Sig-1R (Sig-1R-*E102Q*-YFP) may have an altered ability to bind Biotin-$(G4C2)_{10}$-RNA. Results showed that this mutant of Sig-1R shows a reduced affinity for the $(G4C2)_{10}$-RNA (Fig. 4d; lane 5, wild type; lane 6, mutant).

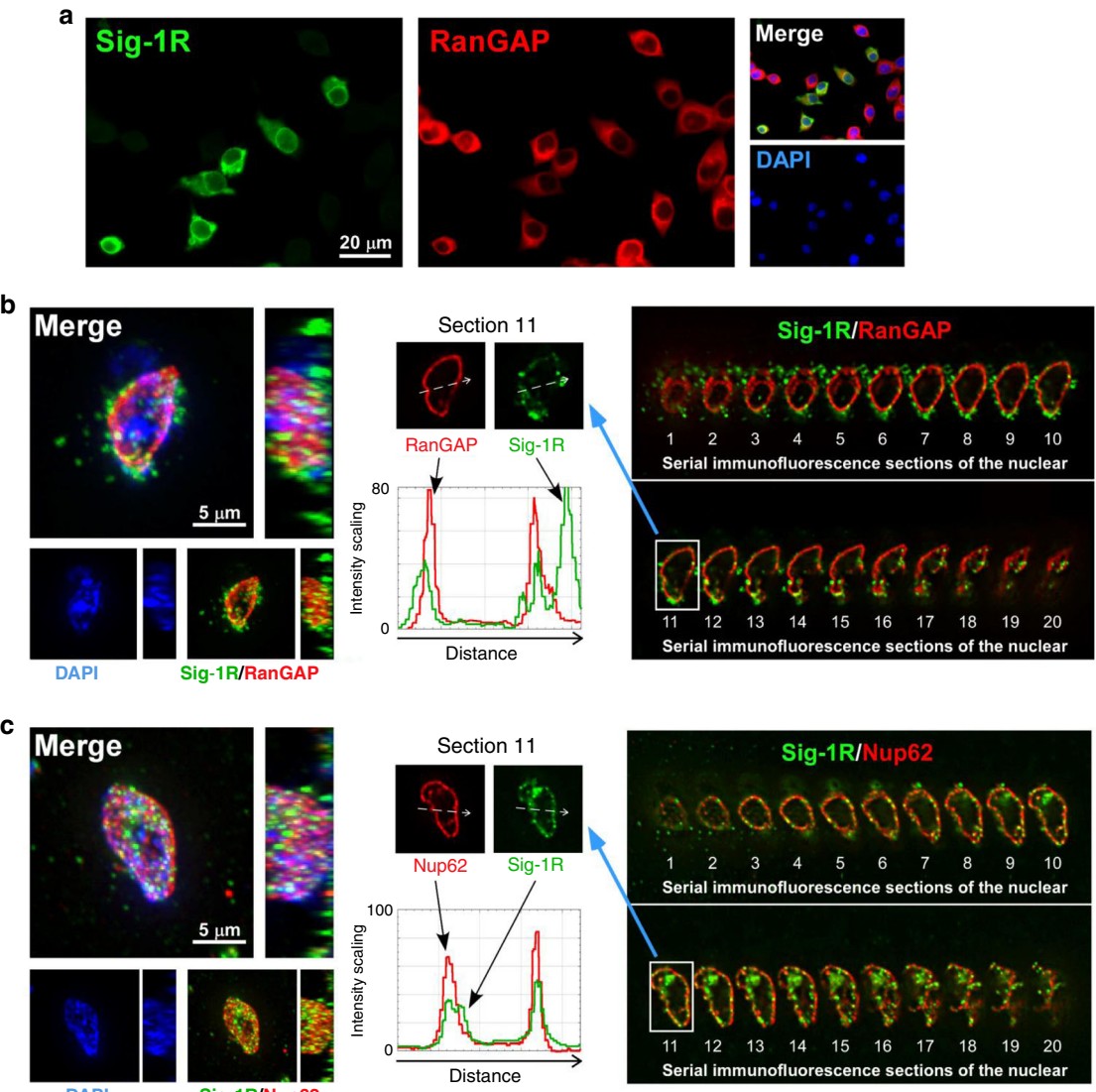

**Fig. 2 Colocalization of HA-tagged Sig-1R, RanGAP, and Nup62 in differentiated NSC-34 motoneuron-like cells. a** Colocalization of HA-Sig-1R with RanGAP. Cells were transiently transfected with pcDNA-HA-Sig-1R, using Lipofectamine 2000, which provided ~50% of transfection efficiency in NSC-34 cells (https://www.thermofisher.com). Two days after transfection, cells were double-labeled with anti-HA and anti-RanGAP antibodies and examined by confocal microscopy. HA-Sig-1R, green; endogenous RanGAP, red; DNA, blue. **b**, **c** Multiple focal planes (Z sections) of the whole nuclear volume were acquired by the DeltaVision microscopy imaging systems. Results of the whole-nucleus image analysis of Sig-1R and RanGAP (**b**) or Nup62 (**c**) are shown. On left panels of **b** and **c**, square images are the top-down view (z-axis), and rectangle panels are a side view (x-axis) of the 3D reconstruction of images. On right panels of **b** and **c**, 20 sections were obtained from a cell. Number 1 is the Z-start at the top surface of the cellular nucleus; number 20 is Z-end at the bottom layer of the nucleus which was near the attachment of the cell to the coverslip. On central panels, white dotted arrows in images of the two sections 11 indicate the track of fluorescence intensity profiles (ImageJ: Plot Profile command) along the arrows. A shift of each focal plane in the Z-axis is 0.25 μm. Three-dimensional reconstructions were made from the Z-series images. Again: HA-Sig-1R, green; endogenous RanGAP/Nup62, red; DNA, blue. Sig-1R, Sigma-1 receptor, RanGAP RanGTP-activating protein, Nup62 nucleoporin 62. $n = 3$ independent experiments with similar results from biologically independent cells.

Thus, the Sig-1R can bind directly the $(G4C2)_{31}$-RNA. The amino acid 102 of Sig-1R seems to play an important role in the interaction.

**Sig-1R effects on (G4C2)-RNA repeats-induced Ran gradient across nuclear membrane in HeLa cells.** Since Sig-1Rs exist at the nuclear pore close to RanGAP (Figs. 1a and 2b) and are, as shown above, able to bind (G4C2)-RNA repeats, the possibility exists that Sig-1Rs might absorb away some of the toxic (G4C2)-RNA repeats from RanGAP in a manner like a molecular sponge. We examined therefore in this section if Sig-1Rs may affect the aberrant nucleocytoplasmic Ran gradient imposed by (G4C2)-

RNA repeats in HeLa cells. Two studies were carried out in this section as follows, i.e., immunocytochemistry and subcellular fractionation followed by western blot.

The $(G4C2)_{31}$-RNA[5] was used to transfect cells. After the transfection, cells were examined by immunocytochemistry for the nucleocytoplasmic gradient (N/C ratio) of Ran. In $(G4C2)_{31}$-RNA-transfected cells, immunoreactive Ran increases significantly in the cytoplasm when compared to controls (Fig. 5).

We next examined if the knockdown of Sig-1Rs might affect the pattern of immunoreactive Ran under the influence of $(G4C2)_{31}$-RNA. Results showed that there is apparently an increase of cytoplasmic Ran in cells treated with shSig-1R when

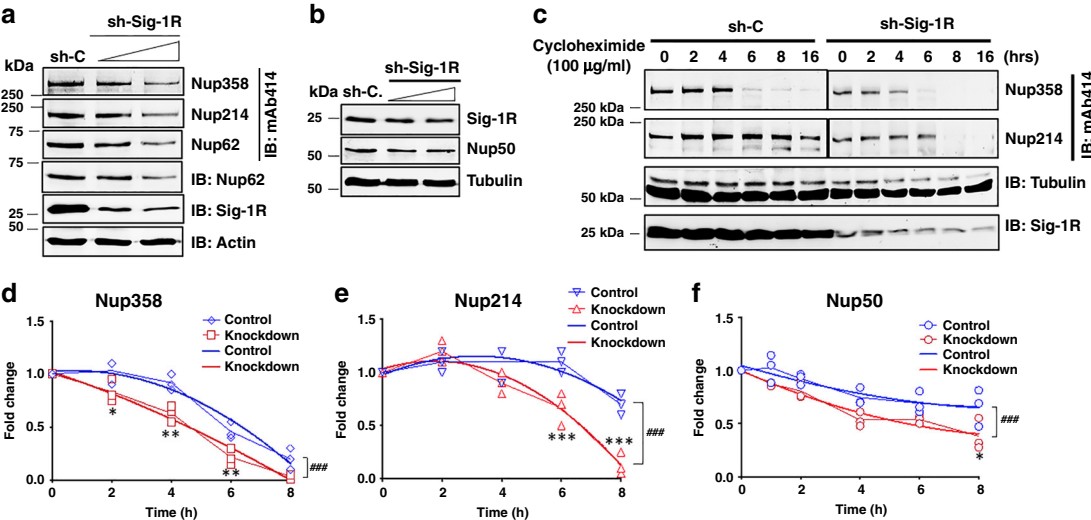

**Fig. 3 Sig-1R regulates nuclear pore protein stability. a** Knockdown of sigma-1 receptor (Sig-1R) by shRNA (shSig-1R) dose-dependently caused a reduction of nucleoporins (Nups). HeLa cells were transiently transfected with either shRNA control (sh-C) or different doses of shSig1R vector. Forty-eight hours after transfection, western blot was performed to detect protein expression levels of Sig-1R, Actin, and Nups. In the western blot, Nup62 was probed by mAb414 or a specific Nup62 antibody. **b** Sig-1R knockdown dose-dependently reduced the level of Nup50. **c** Faster turnover of Nup358 or Nup214 in Sig-1R knockdown cells. Cycloheximide (100 μg/ml) was added to stop de novo synthesis of proteins. Time-lapsed levels of Nups were examined by western blot probed by mAb414. **d–f** Summarized results from three sets of independent turnover studies on Nup358, Nup214, and Nup50. Data were subjected to two-way ANOVA followed by Sidak's multiple comparisons test (Graphpad Prism version 8.2). For **d** Nup358, p values are 0.022, 0.001, 0.0045, and 0.0801 for 2 h, 4 h, 6 h, and 8 h, respectively; for **e** Nup214, p values are 0.7858, 0.1538, 0.0004, and <0.0001 for 2 h, 4 h, 6 h, and 8 h, respectively; For **f** Nup50, p values are 0.2869, 0.583, 0.0602, 0.2899, and 0.0074 for 1 h, 2 h, 4 h, 6 h, and 8 h, respectively. *$p < 0.05$; **$p < 0.01$; ***$p < 0.001$. Data were also subjected to non-linear regression for best fit. For **d** Nup358, $F_{(3,24)}$ value: 11.47; for **e** Nup214, $F_{(3, 24)}$ value: 20.21; for **f** Nup50, $F_{(3, 30)}$ value: 9.754; ###$p < 0.001$. Blue lines: non-linear regression curve for wild-type cells. Sig-1R, Sigma-1 receptor, Nup nucleoporin. $n = 3$ (**a**), $n = 2$ (**b**), and $n = 3$ independent experiments with similar results from biologically independent preparations. Note: data from **c** were analyzed to yield results for **d–f**.

compared to the control shRNA-treated cells (Fig. 6a, b). Successful knockdown of Sig-1Rs by transfection of shSig-1R is shown in a western blot (Fig. 6c).

Subcellular fractionation followed by western blotting was then carried out to quantitatively confirm the immunocytochemistry results. Specifically, the effect of Sig-1R knockout or over-expression on the N/C ratio of Ran was examined. Because the total tubulin level was the same in either wild type or knockout cells (Fig. 7a, lanes 1–4), the N/C ratio of protein of interest was calculated directly by the densitometric ratio from western blot. Results are presented in the rest of this section with the knockout results presented first. Sig-1Rs were knocked out in HeLa cells by the CRISPR technology.

Visual examination on the representative western blot showed that in wild-type cells the cytoplasmic level of Ran apparently is not affected by the (G4C2)31-RNA treatment (Fig. 7a; lane 5 and lane 6) whereas the nuclear level of Ran is reduced by the same treatment (Fig. 7a, lane 9 and lane 10). In knockout cells, while the level of cytoplasmic Ran does not appear to differ between the control and the (G4C2)31-RNA-transfected cells (Fig. 7a; lane 7 and lane 8), the level of nuclear Ran apparently decreases to a large extent (Fig. 7a; lane 11 and lane 12). Interestingly, in wild-type cells, (G4C2)31-RNA causes a decrease of Sig-1R in the cytoplasmic extract (Fig. 7a; lane 5 vs lane 6) while concomitantly causes an apparently slight increase in the nuclear extract (Fig. 7a; lane 9 vs lane 10).

We examined next on the effect of the overexpression of Sig-1Rs on the cytoplasmic and nuclear levels of Ran in wild-type cells treated with (G4C2)31-RNA. Visual examination showed that, at 1 μg or 3 μg of the Sig-1R-YFP gene used for transfection, the expressed level of Sig-1R-YFP did not apparently differ in transfected cells (Fig. 7b; lane 2 and lane 3), suggesting a near

maximum transfection at 1 μg of the vector employed. Results on the Ran level show that the cytoplasmic Ran decreases while the nuclear Ran increases with the overexpression of the Sig-1R-YFP (Fig. 7c).

The N/C ratio of Ran was then quantified by comparing results from three sets of independent western blotting experiments. Results are shown as follows.

In wild-type cells, the (G4C2)31-RNA-transfection significantly causes an increase of cytoplasmic Ran, thus a decrease in the N/C ratio (Fig. 7d). In Sig-1R-knockout cells, however, the N/C ratio of Ran apparently decreases in a greater magnitude when compared to that seen in wild-type cells (Fig. 7d). Those results suggest an even higher cytoplasmic accumulation of Ran in the presence of (G4C2)31-RNA when Sig-1Rs are reduced in the cell.

Overexpression of Sig-1Rs in wild-type cells causes an increase of the N/C ratio of Ran (Fig. 7e). This suggests that the overexpression of Sig-1Rs increases the nuclear Ran, counter-acting the insult of (G4C2)31-RNA that causes the pathological accumulation of cytoplasmic Ran.

On a separate note, it is interesting to notice that in wild-type cells the (G4C2)31-RNA treatment causes an increase of N/C ratio of Sig-1Rs (Fig. 7f), suggesting a translocation of Sig-1Rs from cytoplasm into the nucleus after the treatment of (G4C2)31-RNA.

We extend our biochemical and cellular biological findings above to animals by using *Drosophila* as a model for ALS/FTD.

**Drosophila studies.** The fruit fly *Drosophila melanogaster* has proven to be a powerful model organism to study how HRE causes ALS and FTD neuropathologies[51]. Expression of expanded (G4C2)-RNA repeats in *Drosophila* leads to retinal degeneration, functional deterioration of motor neurons and locomotor defects[52–54]. Among pathogenic mechanisms, *Drosophila* genetics

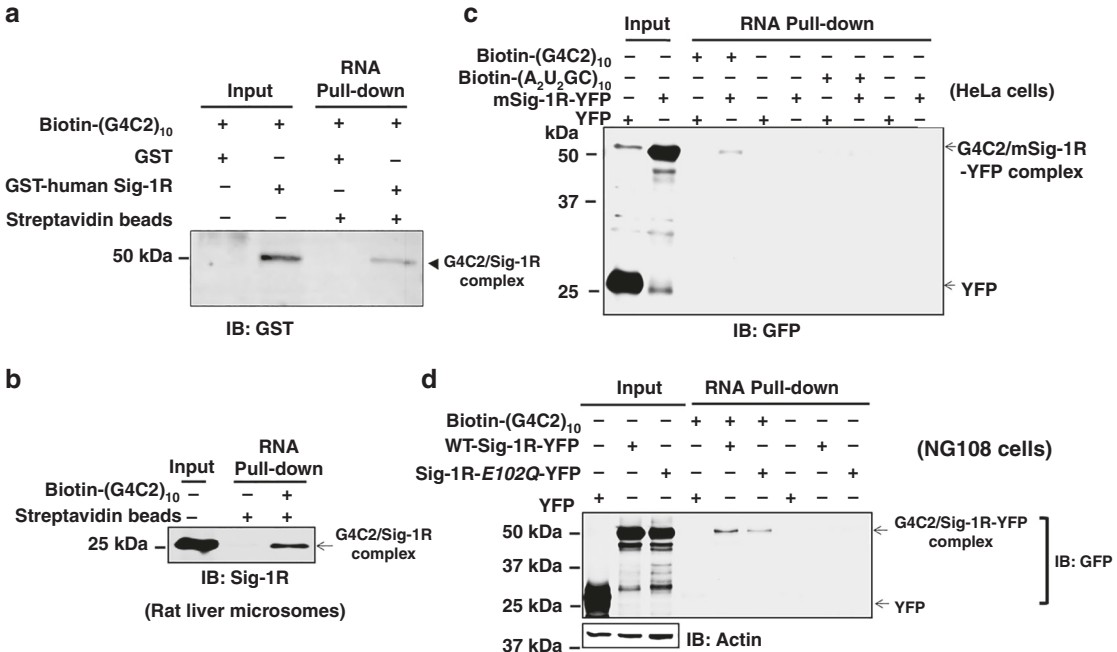

**Fig. 4 Human Sig-1R interaction with (G4C2)₁₀-RNA: direct binding assay with purified molecules in the in vitro cellular assay. a** Purified human sigma-1 receptor (Sig-1R) directly binds (G4C2)₁₀-RNA in a chemical reaction in test tube. The biotin pull-down assay plus western-blot analysis were performed to detect the association of biotin-labeled (G₄C₂)₁₀-RNA and GST-tagged human Sig-1R. **b** (G4C2)₁₀-RNA binds the endogenous Sig-1R in rat liver microsomal preparation. Biotin-labeled (G₄C₂)₁₀-RNA was incubated with lysates from rat liver microsomes followed by the biotin pull-down assay plus western-blot to detect the association of biotin-labeled (G₄C₂)₁₀-RNA and endogenous Sig-1R. **c** Scrambled RNA repeats failed to bind Sig-1R. Biotin-labeled (A₂U₂GC)₁₀-RNA was chosen as the scrambled RNA control. HeLa cells were transfected with mouse Sig-1R-YFP (mSig-1R-YFP) vectors for 24 h before the biotin pull-down assay plus western blot. Note: compare lane 4 vs lane 8. **d** Sig-1R mutation from glutamic acid to glutamine at amino acid 102 (i.e., Sig-1R-E102Q) has a lower affinity for (G4C2)₁₀-RNA. NG-108 cells were transiently transfected with wild-type human Sig-1R-YFP or Sig-1R-E102Q-YFP vector for 24 h before the biotin pull-down assay plus western blot. Note: compare lane 5 vs lane 6. Sig-1R, Sigma-1 receptor. $n = 4$ (**a**), $n = 3$ (**b**), $n = 3$ (**c**), and $n = 3$ (**d**) independent experiments with similar results from biologically independent cellular preparations.

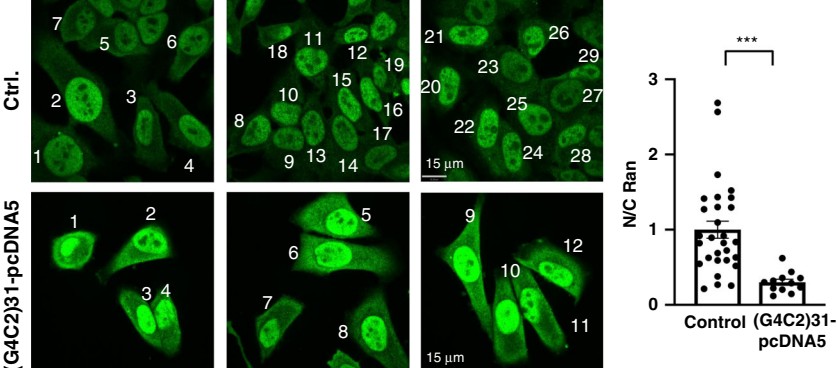

**Fig. 5 (G4C2)31-RNA increases Ran accumulation in the cytosol of HeLa cells: imaging analyses.** The (G₄C₂)₃₁-pcDNA5 or empty vector (Ctrl.) was transiently transfected into HeLa cells. Distribution of endogenous ras-related nuclear protein (Ran; Green) was detected by immunostaining and confocal microscopy. The semi-quantification of cytosolic or nuclear Ran was performed by using NIH Image J. (version 1.51b; right panel). Data are presented as means ± SEM for control cells ($n = 29$) and for cells ($n = 12$) overexpressing (G₄C₂)₃₁-RNA. Two-tailed unpaired Student's $t$ test, $p = 0.0003$, ***$p < 0.001$. Ran: ras-related nuclear protein. $n = 3$ independent experiments with similar results from biologically independent cells.

brought to light the importance of nucleocytoplasmic transport. Enhancing nuclear import was indeed found potent rescuer of HRE-induced toxicity in fly eyes[12,54,55].

To evaluate in vivo whether or not human Sig-1R confers protection against expanded (G4C2)-RNA repeats, we used *Drosophila* (female) models expressing 3 or 30 RNA repeats of G4C2 ((G4C2)₃ and (G4C2)₃₀, respectively) under the regulation of UAS-GAL4 system[52]. We first confirmed that human Sig-1R is properly expressed in the presence of expanded (G4C2; Fig. 8a). Note that *Drosophila* has no detectable Sig-1R (Fig. 8a, left 2

lanes). *Drosophila* eyes are commonly used to evaluate toxicity of genes and in accordance to previous studies[52], expression of (G4C2)₃₀ into the retina progressively induces the formation of degenerative eyes with necrotic spots (Fig. 8b). Of interest, while this phenotype has an incomplete penetrance of 38%, the co-expression of Sig-1R significantly reduces the penetrance to 3% (Fig. 8c).

Flies expressing (G4C2)₃₀ in neurons was also previously found to exhibit reduced locomotor activity[52]. We examined here the climbing response of flies after being tapped in the negative

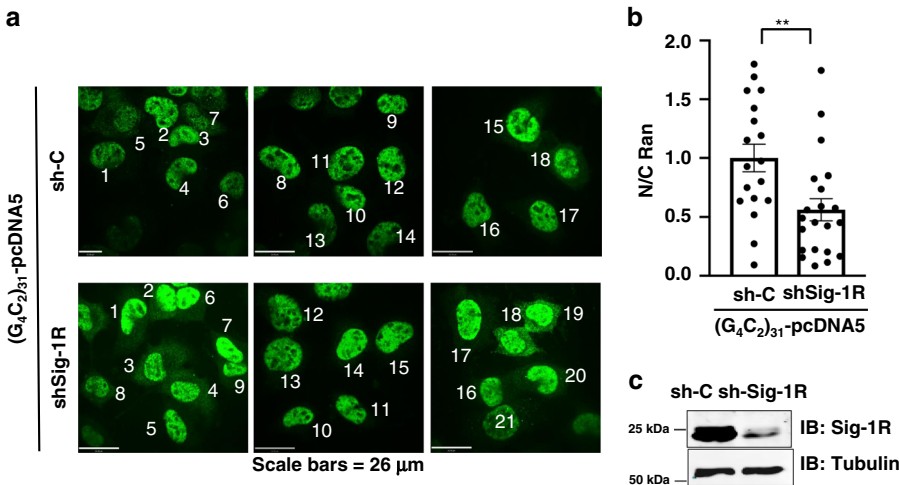

**Fig. 6 Sig-1R knockdown exacerbates the increase of cytoplasmic Ran caused by (G4C2)31-RNA: Imaging analyses. a** The shRNA control (sh-C) or Sig-1R-shRNA (shSig-1R) vector was transiently transfected into HeLa cells for 24 h. The $(G_4C_2)_{31}$-RNA vector was then transfected into HeLa cells to produce the $(G_4C_2)_{31}$-RNA in the cell. Distribution of endogenous Ran (in green) was detected by immunostaining and confocal microscopy. **b** The semi-quantification of cytosolic or nuclear Ran was performed by using NIH Image J. (version 1.51b). Data are presented as means ± SEM; $n = 18$ for biologically independent control cells receiving scrambled shRNA plus $(G_4C_2)_{31}$-RNA; $n = 21$ for biologically independent cells receiving shSig-1R plus $(G_4C_2)_{31}$-RNA. Two-tailed unpaired Student's $t$ test, $p = 0.0056$. **$p < 0.01$. **c** The knockdown efficiency of Sig-1R protein in experiment above was detected by western blot. In this experiment, shSig-1R was transfected twice, i.e., 24 h after the first transfection the second transfection was carried out. Sig-1R, Sigma-1 receptor; Ran, ras-related nuclear protein. $n = 3$ (**a**) and $n = 2$ (**b**) independent experiments with similar results from biologically independent cells or cellular preparations.

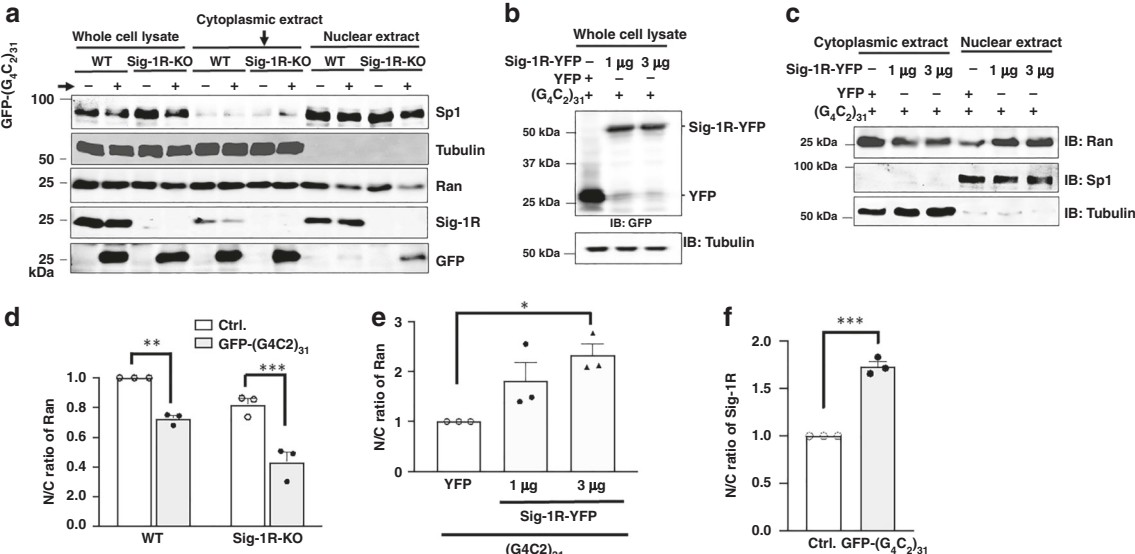

**Fig. 7 Sig-1R knockout exacerbates whereas Sig-1R overexpression attenuates the cytoplasmic Ran accumulation caused by (G4C2)31-RNA: western blot analyses. a** Apparent decrease of nuclear Ran in Sig-1R knockout cells when compared to that seen in wild-type cells. Subcellular fractionation followed by western blot was used to examine the level of Ran in the cytoplasm or nucleus in HeLa cells in which Sig-1Rs were depleted by the CRISPR/Cas9 technique. GFP-$(G_4C_2)_{31}$ vectors were transiently transfected for 24 h into either wild-type or CRISPR/Cas9 Sig1R-KO cells before subcellular fractionation. **b** Sig-1R successfully overexpressed in (G4C2)31-RNA-treated cells. **c** Overexpression of Sig-1R decreased Ran in the cytoplasm while concomitantly increased Ran in the nucleus. **d** Sig-1R knockout apparently exacerbates the N/C ratio of Ran when compared to that seen in wild type. See Results for explanations. Quantitative summary of results from three sets of independent experiments illustrated in **a**. Data are presented as means ± SEM; $n = 3$ for each group; two-way ANOVA followed by Sidak's multiple comparisons test, $p = 0.0027$ for wild-type group, $p = 0.0003$ for Sig-1R-KO group, **$p < 0.01$; ***$p < 0.001$. **e** Overexpression of Sig-1Rs significantly rescues the aberrant N/C ratio of Ran caused by the $(G4C2)_{31}$-RNA. Quantitative summary of results from three sets of independent experiments illustrated in **c**. Data are presented as means ± SEM; $n = 3$ for each group; one-way ANOVA followed by Sidak's multiple comparisons test, $p = 0.172$ for YFP vs 1 μg, $p = 0.0276$ for YFP vs 3 μg, $p = 0.4723$ for 1 μg vs 3 μg, *$p < 0.05$. **f** Increase of nuclear Sig-1Rs in the (G4C2)31-RNA-treated HeLa cells. Quantitative summary of results from three sets of independent experiments illustrated in **a**. Data are presented as means ± SEM; $n = 3$ in each group. Two-tailed unpaired Student's $t$ test ($p = 0.0001$), ***$p < 0.001$. Note: the N/C ratio of the control in each independent experiment in **d–f** was taken as one. The N/C ratio from each experiment was normalized to respective control. Sig-1R: Sigma-1 receptor; Ran: ras-related nuclear protein. $n = 3$ (**a**), $n = 3$ (**b**), and $n = 3$ (**c**) independent experiments with similar results from biologically independent cellular preparations.

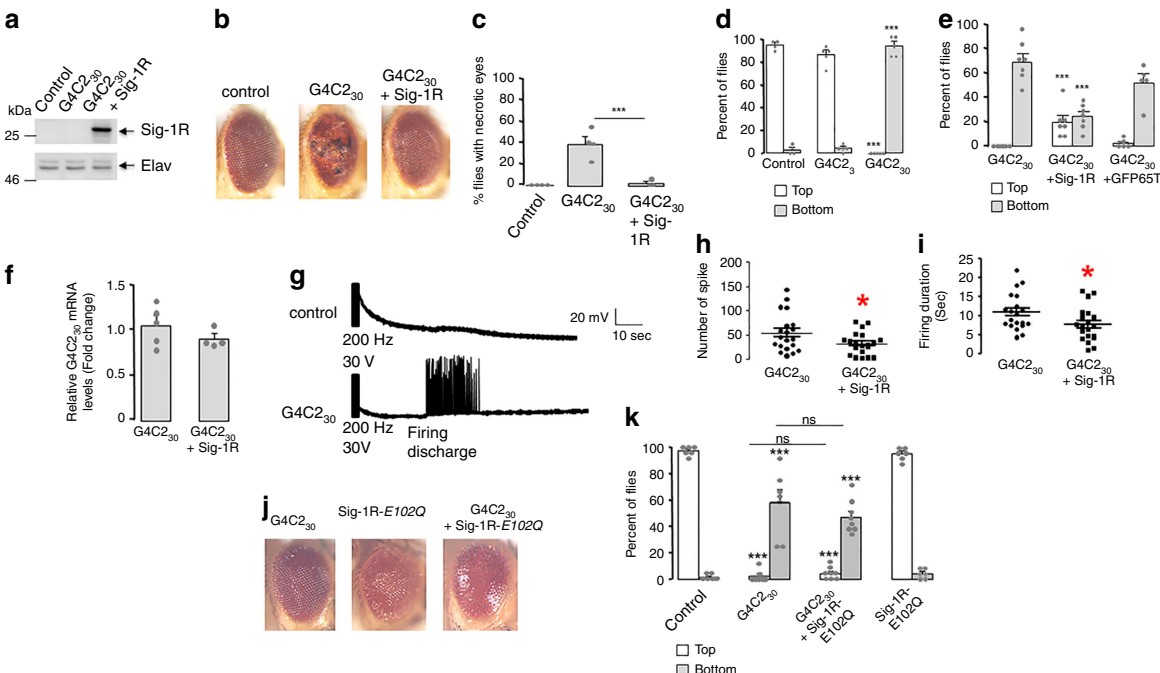

**Fig. 8 Sig-1R, but not the Sig-1R-E102Q mutant, rescues the phenotype of flies expressing expanded (G4C2) repeats. a** Western blot of human Sig-1R (Sigma-1 receptor) in *Drosophila*; $n = 3$ independent experiments with similar results from biologically independent preparations. **b** Representative external eye morphology of flies (20 days after eclosion) expressing no transgene (control), 30 G4C2 repeats ($(G4C2)_{30}$) alone or together with human Sig-1R. **c** Quantification of flies presenting necrotic spots in the eyes; $n = 4$ independent sets of studies from a total of 91 flies in control, 48 in $(G4C2)_{30}$, and 68 in $(G4C2)_{30} + $ Sig-1R group; statistics at the end. **d** Climbing performances (observation starts at 1 min) of 4-day-old flies expressing no transgene (control), 3 ($(G4C2)_{3}$) or 30 G4C2 repeats ($(G4C2)_{30}$). $n = 8$ flies/group; number of trials: control, 4; $(G4C2)_{3}$, 5; $(G4C2)_{30}$, 5. ***$p < 0.001$ versus control; statistics at the end. **e** Climbing performances of 4-day-old flies expressing $(G4C2)_{30}$ alone or with Sig-1R or the green fluorescent protein GFP (GFP65T) in neurons. $n = 8$ flies/group; number of trials: $(G4C2)_{30}$, 7; $(G4C2)_{30} + $ Sig-1R, 7; $(G4C2)_{30} + $ GFP65T, 5; statistics at the end. **f** Sig-1R does not modify G4C2$_{30}$ mRNA expression. mRNA levels of G4C2$_{30}$ ($n = 5$) or $(G4C2)_{30} + $ Sig-1R ($n = 4$) were normalized to actin. Data are means ± S.E.M.; unpaired two-tailed *t*-test ($p = 0.2950$). **g** Representative traces of evoked responses after an electroconvulsive stimulation (30 V, 200 Hz) in flies expressing no transgene (control) or $(G4C2)_{30}$. **h** Number of spikes in firing discharges induced by an electroconvulsive stimulation on flies ($n = 20$/group) expressing $(G4C2)_{30}$ alone or together with Sig-1R. Data are means ± S.E.M; unpaired two-tailed *t*-test (*$p = 0.0385$). **i** Duration of firing discharges for flies ($n = 20$/group) expressing $(G4C2)_{30}$ alone or together with Sig-1R. Data are means ± S.E.M.; unpaired two-tailed *t*-test (*$p = 0.0375$). **j** Expression of Sig-1R-E102Q in retina leads to rough eye phenotype. **k** Sig-1R-E102Q failed to ameliorate climbing performances of flies (4 days old) expressing expanded G4C2 repeats. Control: no transgene. $n = 8$ flies per group; number of trials: Control, 6; $(G4C2)_{30}$, 7; $(G4C2)_{30} + $ Sig-1R- E102Q, 8; Sig-1R- E102Q, 6. Statistics for **c**, **d**, **e**, **k**: data are means ± S.E.M., one-way ANOVA plus Tukey's multiple-comparison tests, ***$p < 0.001$ versus control, ns not significant.

geotaxis test. This locomotor test uses the natural reflex of flies to walk against gravity and is a standard locomotor activity test in the field of *Drosophila*. Expression of $(G4C2)_{30}$ but not $(G4C2)_{3}$ in neurons, under the control of Elav-GAL4 driver, led to strong climbing response defects (Fig. 8d). While >80% of control or $(G4C2)_{3}$-expressing flies attain the top of the column within 1 min, most of the $(G4C2)_{30}$-expressing flies do not climb and none of them succeed to reach the top. Indeed, most of $(G4C2)_{30}$-expressing flies suffered from a seizure-like episode after being startled. The bang-sensitive behavior was previously described as an abnormal response of the giant fiber escape circuit that controls motor neurons[56,57]. The presence of Sig-1R significantly ameliorates the climbing deficit of G4C2$_{30}$-expressing flies. In this case, 76% of flies climbed along the column and 19% of them reached the top within 1 min (Fig. 8e). As a control for potential UAS-GAL4 dilution effect, overexpressing the green fluorescent protein (GFP65T) fails to modify climbing performances of G4C2$_{30}$-expressing flies (Fig. 8e). Moreover Sig-1R does not modify G4C2$_{30}$ RNA expression (Fig. 8f), indicating that Sig-1R acts downstream of $(G4C2)_{30}$ transcription.

Electrophysiological studies previously showed that the bang-sensitivity correlates to long firing discharge at the neuromuscular junction after a high-frequency electroconvulsive stimulation of the giant fiber pathway[56,57]. We thus set up electrophysiological

recordings of flight muscles after stimulation (200 Hz for 2 s) of giant fiber neurons into the brain. As expected, an electroconvulsive stimulation of $(G4C2)_{30}$-expressing flies resulted in delayed firing discharge in flight dorsal muscles (Fig. 8g). The presence of Sig-1R decreases the number of spike but also the duration of firing discharge (Fig. 8h, i). Thus, we demonstrate that the Sig-1R significantly reduces abnormal long firing of giant fibers and thereby ameliorates the startle-induced climbing response of flies expressing $(G4C2)_{30}$.

We next examined whether the Sig-1R-E102Q mutant[41,43] may have an impact on the $(G4C2)_{30}$-induced phenotype. The Sig-1R-E102Q was either expressed in eyes only or in neurons, respectively, for examination of eye morphology or locomotion.

While flies expressing Sig-1R-E102Q in eyes only under the regulation of GMR-GAL4 driver shows a mild rough eye phenotype at 1 day of age, the presence of $(G4C2)_{30}$ seems to worsen this phenotype (Fig. 8j). Unexpectedly, we also found that co-expression of $(G4C2)_{30}$ with Sig-1R-E102Q in eyes is lethal for adult flies. They only survive a few days after eclosion (3–10 days), thus hampering a statistical analysis of the degenerative eye phenotype when they developed and aged.

We previously showed that mutant Sig-1R-E102Q has no deleterious locomotor effects in *Drosophila* notably when expressed at a moderate level in neurons such as in the line of

Sig-1R-$E102Q$#1 flies[43]. Here we used the same Sig-1R-$E102Q$#1 flies for this portion of the study. Accordingly, Sig-1R$E102Q$#1 shows no climbing deficit when compared to control (Fig. 8; far right panels). When $(G4C2)_{30}$ was expressed in neurons of Sig-1R$E102Q$#1 using the Elav-GAL4 driver, flies now present climbing defects seen in $(G4C2)_{30}$-expressing flies at 4 days of age (Fig. 8k). Altogether, these data suggest that the mutant Sig-1R-$E102Q$ confers no protection against $(G4C2)_{30}$ toxicity.

## Discussion

The nuclear pore complex (NPC) has been coined "The gate to neurodegenerative diseases"[58]. Here we report the existence of the first molecular chaperone at the NPC and show that this chaperone, the Sig-1R, counteracts the N/C ratio deficit of Ran induced by the (G4C2)-RNA repeats that underlies ~40% of the familial ALS cases. Although this study used cellular models in the first part of the study, the potential implications of results should not be lightly discounted for the following reasons. Firstly, we used a human cell line here in the present study. Secondly, we have shown in the past that results from cell lines perfectly mimic the results from rodent brain[39,59]. In addition, we have used *Drosophila* in this study which has been recognized as a suitable model for ALS and FTD and demonstrated that almost all biochemical and cellular biological observations in HeLa cells can be validated by the animal study.

Of course, more studies need to be done in the future to examine the clinical implication of the current report. Nevertheless, the direct implication of the current study is that by increasing Sig-1Rs in the ALS/FTD patients, suffering from the insult of the (G4C2)-RNA repeats, may attenuate the damage caused by the RNA repeats. In this regard, it is interesting to note that at least three drugs have been shown to increase Sig-1Rs in cell cultures and in rodents[60–63]. Follow-up studies in this line of thought may lead to potential therapeutic agents for treatment of this type of ALS/FTD patients.

Our study is not the first to relate Sig-1Rs to ALS/FTD but is the first to point out the NPC as the site of action of Sig-R in this regard. It is interesting to note that Sig1-Rs exist in diverse places in a cell including the nuclear envelope[38,39] and nucleoplasm[38]. This may explain why Sig-1Rs may affect the stability of Nups that are on the cytosolic side but also other Nups that face the nucleoplasm such as Nup50. Whether Sig-1Rs may chaperone some other Nups not examined in this study is unknown. Since the crystal structure of NPC is known[64–66], it would be interesting to know where and how this one-transmembrane Sig-1R[25] fits into the structure of NPC. It is not totally clear at present how the chaperoning activity of Sig-1R may, at the molecular level, affect the N/C ratio of Ran. We surmise that the stabilized Nups, thus the NPC assembly, may facilitate the Ran entry from cytosol into nucleus.

The interaction between Sig-1R and $(G4C2)_{10}$-RNA was discovered in this study out of the extension of the Sig-1R's known ability in binding proteins as well as lipids[49,50]. The results of this study thus place the Sig-1R as an RNA-binding protein (i.e., ribonucleoprotein). Other ribonucleoproteins, including TDP-43[67] and FUS[68], are also known to involve in the neurodegenerative disease. Several studies have reported an increase of Sig-1Rs in the nucleus of neurons related to several neurodegenerative diseases (e.g. refs. [40,41,69,70]). Here we also see an increase of Sig-1Rs in the nucleus of $(G4C2)_{31}$-RNA-treated cells (Fig. 7f). We do not know at present why Sig-1Rs are increased in the nucleus of those "diseased" cells exactly opposite to that seen with dysfunctional TDP-43 or FUS. The relation, if any, between those three critical ribonuclear proteins in neurodegeneration remains to be cleared in the future.

We speculate that the Sig-1R's rescue of the Ran N/C ratio may result from the Sig-1R ability to bind (G4C2)-RNA repeats as a molecular sponge and to reduce thus the effective concentration of the RNA repeats as they exert their insults on RanGAP[12]. It is interesting to note that the Sig-1R is in close proximity to RanGAP (Figs. 1a and 2a, b) and in fact can co-IP with RanGAP (Fig.1b). Thus, it can be imagined that Sig-1R-RanGAP-$(G4C2)_n$-RNA may exist as a trimeric complex. If so, how does the Sig-1R help the RanGAP to get rid of the toxic $(G4C2)_n$-RNA? More studies are certainly warranted to provide answer to this question.

Although we show here that the Sig-1R with a single amino acid mutation at 102 has a reduced ability to bind $(G4C2)_{10}$-RNA (Fig. 4d), we do not know if this mutation of Sig-1Rs plays a role in the (G4C2)-RNA repeats-induced ALS/FTD. It rarely happens that a disease is caused by two mutations. Nevertheless, our result suggests a structural specificity of Sig-1R in its interaction with (G4C2)-RNA repeats. Whether the Sig-1R can interact with the HRE is unknown at present.

Questions as mentioned above notwithstanding, our results suggest the Sig-1R as a never-before reported target in understanding the NPC- and (G4C2)-RNA repeats-related neurodegeneration. The Sig-1R has been implicated as a beneficial factor in many types of neurodegenerative diseases in part due to the receptor's ability to regulate the downstream targets at multiple loci of a cell[35]. Our current result indicates yet another locus whereby the Sig-1R plays a role against the neurodegenerative disease. In this case it is against the C9orf72 type of ALS/FTD at the NPC. Since the nuclear pore has been indicated to involve in many neurodegenerative diseases[58], it is tempting to suggest that the Sig-1R action at the nuclear pore may serve as a common molecular target for those diseases.

Nevertheless, other potential mechanisms or loci may also be involved in the (G4C2)-RNA repeats-antagonizing action of Sig-1Rs. For example, the HRE-derived dipeptide repeats (DPR), can impede the maturation of mRNA in the nucleus or the biogenesis of ribosomal RNA in nucleoli, in model cells and even in patient's cells, leading to defective nucleocytoplasmic transport[71]. Inasmuch as the Sig-1R exists in the nucleus, albeit with its nucleolar presence yet to be determined, it is tempting to speculate that Sig-1Rs may regulate the maturation of mRNAs or biogenesis of ribosomal RNA. Of course, whether Sig-1Rs may directly bind the DPR or the small RNA of the component of RNA spliceosome remains to be determined. The action of DPR was also reported to relate to the formation of stress granule assembly in the cytosol that plays a critical role in impeding the nucleocytoplasmic transport[10]. As Sig-1Rs exist at the reticular network of ER directly facing cytosol, they may participate in the formation of the granule assembly.

Existing evidence suggests a relation between the Sig-1R and TDP-43. For example, overexpression of TDP-43 causes a locomotor deficit as well as a reduced production of ATP in *Drosophila*, both of which nonetheless are rescued by overexpression of Sig-1R[43]. Those results suggest an action of TDP-43 at the mitochondria or perhaps at the IP3R of the MAM where the Sig-1R functions as a chaperone. TDP-43 pathology is recently shown to relate clinically to (G4C2)-RNA repeats[72]. Taken together, those results indirectly suggest a mitochondrial energy metabolism deficit caused by (G4C2)-RNA repeats, perhaps through TDP-43 which then is counteracted by the Sig-1R.

Lastly, it is interesting to see that the overexpression of Sig-1R-$E102Q$ per se causes morphological deficit in the eye (Fig. 8j). In fact, this gain-of-toxicity of Sig-1R-$E102Q$ may be related to its action on the IP3R at the MAM important for ATP production[43]. This explanation on the action of the Sig-1R-$E102Q$ at IP3R at the MAM, together with current result showing the $(G4C2)_{30}$ co-expression exacerbating the toxic effect of the mutant in the eye

(Fig. 8j), render support to the above notion that (G4C2)-RNA repeats may lead to dysfunctional mitochondria and a reduced cellular bioenergetics. Further study is certainly required to confirm this speculation.

## Methods

**Sig-1R knockout cells: selection on HeLa cells.** HeLa human cervical cancer cells and mouse neuroblastoma × rat glioma cells were purchased from American Type Cell Collection (ATCC). HeLa cells were maintained in Dulbecco's modified Eagle's medium (GIBCO) supplemented with penicillin (100 units/mL), streptomycin (100 μg/mL), and 10% Fetalgro bovine growth serum (RMBIO). Human Sig-1R CRISPR/Cas9 knockout (KO) and Sig-1R HDR plasmids (Santa Cruz) were transiently co-transfected into HeLa cells with the Lipofectamine 2000 transfection reagent (Thermo Fisher Scientific) and were cultured in Dulbecco's modified Eagle's medium (GIBCO), penicillin (100 units/mL), streptomycin (100 μg/mL), and 10% Fetalgro bovine growth serum (RMBIO). For the selection of stably transfected Sig-1R CRISPR/Cas9-KO cells, cells were maintained in culture media supplemented with puromycin (100 μg/mL, GIBCO) for stable cell lines selection to generate permanent HeLa-Sig-1R-KO cells. NG-108 cells were cultured in Eagle's minimum essential medium (GIBCO) supplemented with penicillin (100 units/mL), streptomycin (100 μg/mL), and 10% Fetalgro bovine growth serum (RMBIO). All cell lines were kept at 37 °C in a humidified 5% $CO_2$ incubator (Thermo Fisher Scientific).

**NSC-34 cell culture.** The NSC-34 cell line was a kind gift from Yijuang Chern's Laboratory of Taiwan. NSC-34 cells were grown in complete culture Dulbecco's modified Eagle's medium (DMEM) containing 10% fetal bovine serum (FBS) and 1% penicillin/streptomycin. To ensure high quality, after passage 30, the cells were no longer used. For differentiation, NSC-34 cells were induced to differentiate into a motoneuron-like phenotype by differentiation medium (1:1 DMEM/Ham's F12 plus 1% of FBS, 1% of non-essential amino acids, and 1% of penicillin/streptomycin) for 4 days. All materials used in NSC-34 cell culture were purchased from Thermo Fisher Scientific (Waltham, MA, USA). Cell monolayers at 50% confluency were used for transfection with plasmids using PolyJet reagent according to the manufacturer's protocol (SignaGen Laboratories, Gaithersburg, MD, USA). In all, 3 μl of PolyJet was incubated with 1 μg of different plasmids in 0.2 ml of serum-free medium for 30 min at room temperature. Subsequently, the cells in a 3.5-cm dish were transfected by the DNA- PolyJet complexes in 2 ml of differentiation medium, and then incubated at 37 °C in 5% CO2 for 6 h. The cells were then incubated for an additional 36 h using 2 ml of fresh medium. The transfection efficiency was mostly >50%.

**Immunostaining.** *HeLa cells:* Cells were seeded on glass slide with coverslip overnight at 37 °C in an incubator followed by fixation with 4% paraformaldehyde in PBS at 4 °C for 20 min. After washing with PBS three times, cells were incubated with permeabilization buffer (0.4% Triton X-100 in PBS) for 15 min. After washing three times with PBS, slides were incubated with SuperBlock blocking buffer (Thermo Fisher Scientific) at room temperature for 30 min and incubated thereafter with indicated primary antibodies in SuperBlock blocking buffer at proper dilution overnight at 4 °C. Cells were then washed three times with wash buffer (0.1% Triton X-100 in PBS) and incubated in SuperBlock blocking buffer with Alexa488- or Alexa594-conjugated secondary antibodies for 1 h. Cells were then washed three times with PBS and mounted with Prolong gold antifade mountant with DAPI. Images of cells were captured by confocal microscopy (Perkin-Elmer Modular laser system 2.0 with Nikon Eclipse TE2000E microscope and Volocity version 6.3 software).

*NSC-34 cells:* Cells were fixed with 4% (vol/vol) paraformaldehyde (Sigma-Aldrich, St. Louis, MO, USA) in PBS and permeabilized with 0.05% Triton-X 100 in PBS for 3 min. Immunostaining was conducted with primary antibodies such as anti-HA (Proteintech,Chicago, IL, USA), anti-GFP/YFP (Takara Bio/Clontech, Mountain view, CA, USA), anti-Nucleoporin p62/Nup62 (Abcam, Cambridge, UK), or anti-RanGAP1 antibodies (Cell Signaling Technology, Danvers, MA, USA) at room temperature for 1 h. Cells were then treated with Alexa Fluor 488- or Alexa Fluor 568-conjugated second antibodies (Thermo Fisher Scientific). Finally, cells were mounted in 90% (vol/vol) glycerol containing 4′-6-diamidino-2-phenylindole (DAPI, Thermo Fisher Scientific) and examined by using the DeltaVision Spectris Imaging System (Applied Precision) that includes an Olympus X71 microscope and the softWoRx software (version 6). The software deconvolves images to improve contrast, by relocating signal scatter and out-of-focus data, to generate images in the 3D rendering.

**Plasmid constructs and gene silencing.** The primer pairs for specific gene amplification in the polymerase chain reaction (PCR) are listed in Supplemental Information Table S1. Those sequences were designed based on the nucleotide database of the National Center for Biotechnology Information (NCBI) and were purchased from the Integrated DNA Technologies (IDT). The coding sequence (CDS) of human Sig-1R (NCBI accession: NM_005866) was amplified by using PCR from the complementary DNA (cDNA) of HeLa cells. The CDS of mouse Sig-

1R (NCBI accession: NM_011014) was amplified from cDNA of neuro-2a cells. To generate the expressing construct of Sig-1R in mammalian cells and the recombinant protein of Sig-1R from *E. coli*, the PCR products of the human Sig-1R containing two restriction enzyme sites *Eco*RI and *Xho*I (New England Biolabs) were purified by using the Wizard SV Gel and PCR Clean-Up System (Promega). Purified PCR products were then used to perform the pGEM-T Easy vector ligations (Promega), by using the NEB-5 alpha Competent *E. coli.* (New England Biolabs), and sub-cloned from the pGEM-T Easy vector either into pcDNA3-HA to produce the HA-human Sig-1R or into pGEX-6p3 (GE Healthcare) to produce the pGEX-6p3-human Sig-1R constructs. The purified PCR products of the mouse Sig-1R containing two restriction enzyme sites *Eco*RI and *Xho*I were used to perform the T&A cloning (Yeastern Biotech Co. Ltd.) and then sub-cloned from T&A plasmid into pGEX-6p3 to produce the pGEX-6p3-mouse Sig-1R construct.

Site-directed mutagenesis was performed by using the QuickChange site-directed mutagenesis kit (Agilent) to generate human and mouse Sig-1R-E102Q/pGEX-6p3 mutants, respectively. Gene knockdown techniques was performed to downregulate expression levels of the Sig-1R in the cell. The short hairpin RNA (shRNA) against the CDS of Sig-1R, which was cloned into Green Flourscence Protein (GFP)-expressing vector (PLKO.1-hGPK-Puro-CMV-tGFP), and the non-targeting shRNA negative control (MISSION PLKO.1-hGPK-Puro Non-Mammalian shRNA control plasmid DNA) were purchased from Sigma-Aldrich. For the sequence of the Sig-1R shRNA (i.e., shSig-1R), two complementary oligonucleotides were chosen: 5′-*GATCC***ACACGTGGATGGTGGAGTA***TTCAAGAGA*TACTCCACCATCCACGTGTTTTTTTGCTAGCG-3′ and 5′-*AATTC*GCTAGCAAAAAAACACGTGGATGGTGGAGTA*TCTCTTGAA*TACTCCAC**CATCCACGTGT**G-3′, where bold letters stand for rat Sig-1R gene 516-534, italics stand for either *Bam*HI or *Eco*RI overhangs, and the bold italics stand for hairpin loop sequences (Hayashi and Su[73,74]; Tsai and Chuang et al.[39]). Transfections of the shSig-1R were performed following manufacturer's recommendations by using Lipofetamine 2000 (Thermo Fisher Scientific). In some case, the transfection was performed twice (e.g., Fig. 6c). The first transfection was in Opti-MEM (reduced MEM). Six hours later, the culture medium was changed to complete medium. Twenty-four hours later, the transfection was performed one more time.

**Immunoprecipitation.** HeLa cells were harvested in 0.3 mL of IP lysis buffer (50 mM NaCl, 0.5% Nonidet P-40, 10 mM Tris-HCl pH 8.0, and 1× protease inhibitor) for 30 min. Protein amounts were measured (Pierce bicinchoninic acid protein assay kit, Thermo Fisher Scientific, Rockford, IL) after centrifugation (18,407×g for 10 min at 4 °C). Protein lysates (500 μg) were incubated with the target antibody (2 μg) and IP lysis buffer in a total volume of 1000 μl and rotated overnight at 4 °C. The Protein-A/G magnetic beads were pre-washed three times with IP lysis buffer and then added into the lysate/antibody mix and rotated for 1 h at 4 °C. After incubation, beads were washed three times with IP lysis buffer containing protease inhibitors (Roche Diagnostics, Indianapolis, IN) for 5 min at 4 °C. After third washing, beads were eluted with 50 μl SDS 2X sample buffer containing dithiothreitol and heated at 95 °C for 10 min. After elution, proteins samples in the mixture were immediately fractionated by using SDS/PAGE as described below in the western blot section to examine the potential protein interaction.

In the Sig-1R-Nup50 co-IP experiments (Fig. 1d), HeLa cell protein lysates (200 μg) were incubated with Nup50 or control IgG antibody (3 μg) and IP lysis buffer in a total amount of 1000 μl and rotated for 2 h at 4 °C. The pre-cleared protein-A/G Agarose (50 μl; Santa Cruz Biotechnology) was added into the protein lysate/antibody mixture and rotated overnight at 4 °C. Beads were washed three times with IP lysis buffer containing protease inhibitors (Roche Diagnostics, Indianapolis, IN) for 5 min at 4 °C. Each wash was accompanied with a 9391×g centrifugation for 1 min at 4 °C to remove the supernatant. After the third wash, the pellet is eluted with 50 μl SDS 2X sample buffer and heated at 95 °C for 10 min. After elution, proteins samples in the mixture were immediately fractionated by using SDS/PAGE and immunoblotted with Nup50 or Sig-1R antibody overnight at 4 °C to examine for their potential interactions. Membranes were washed three times for 15 min followed by probing with secondary antibody of "peroxidase-conjugated Affinipure goat anti-mouse IgG" for Sig-1R or "peroxidase-conjugated IgG fraction monoclonal mouse anti-rabbit IgG, light-chain specific" for Nup50. Blots were washed three times for 15 min with TBST and developed by using the Azure Biosystem C600.

**Western blot: HeLa cell, NG-108 cells.** Total proteins were extracted from HeLa (ATCC) and NG-108 cells (ATCC). Briefly, collected cells were lysed with RIPA lysis buffer (10 mM Tris-HCl pH 8.0, 140 mM sodium chloride, 0.1% sodium dodecyl sulfate, 1% Triton X-100, 1 mM ethylenediaminetetraacetic acid, and 0.5 mM ethylene glycol tetraacetic acid) containing protease inhibitors (Roche Diagnostics, Indianapolis, IN) and the protein amount was measured (Pierce bicinchoninic acid protein assay kit, Thermo Fisher Scientific, Rockford, IL). Equal amount (30 μg) of protein samples were fractionated by using SDS-polyacrylamide gel electrophoresis (SDS-PAGE) and transferred onto a polyvinylidene difluoride membrane. After incubation with 5% (wt/vol) nonfat milk in TBST (10 mM Tris. pH 8.0, 150 mM NaCl, and 0.5% (vol/vol) Tween 20) for 1 h, membranes were incubated with various primary antibodies (see Supplementary Information Table 1) overnight at 4 °C. Alpha-tubulin or actin was used as loading control.

Membranes were washed three times with TBST for 15 min followed by probing with secondary antibody of goat anti-rabbit or goat anti-mouse antibody. Blots were washed three times for 15 min with TBST and developed by using the LiCor system (LiCor CLx). Expression bands were analyzed by Image Studio Lite (LiCor 5.2.5) according to the manufacturer's manual.

**Nuclear/cytoplasmic fractionation of HeLa cells**. HeLa (or HeLa-Sig-1R-KO) cells were grown to 80% confluency and transiently transfected with indicated vectors including pcDNA5($C_4G_2$)$_{31}$ and pEGFP-N3.31-($C_4G_2$)$_{31}$ (gifts from Mauro Cozzolino). Twenty-four hours after transfection, cells were harvested for sub-cellular fractionation. The subcellular fractionation was performed by using the Subcellular Protein Fraction Kit for Cultured Cells (Thermo Fisher Scientific). The procedure is briefly described per manufacturer's instructions as follows. HeLa cells were rinsed once with phosphate-buffered saline (PBS) and gently lysed with cytoplasmic extraction buffer that was supplemented with protease inhibitor cocktail at 4 °C for 10 min. The cytoplasmic fraction (supernatant) were collected by centrifugation at 500×$g$ for 5 min at 4 °C. The pellets were then incubated with membrane extraction buffer at 4 °C for 10 min and the membrane fraction (supranate) was prepared by centrifugation at 3000×$g$ for 5 min at 4 °C. For the isolation of the nuclear extract, the resultant pellets were subsequently incubated with nuclear extraction buffer at 4 °C for 10 min and the soluble nuclear extract (supernatant) were collected by centrifugation at 5000×$g$ for 5 min at 4 °C.

**Biotin pull-down assay**. The commercially synthesized biotinylated RNAs were purchased from Integrated DNA Technologies (Table S2; Supplementary Information). The purified recombinant proteins, HeLa cell lysates, or extracts from rat liver microsomes (BIOIVT) were incubated with 20 nM biotin-labeled RNAs in binding buffer (10 mM HEPES pH 8.0, 40 mM KCl, 3 mM MgCl$_2$, 5% glycerol, 2 mM DTT, 0.5% Nonidet P-40, and 1% tween-20) supplemented with protease inhibitor cocktail (Roche) and were rotated overnight at 4 °C. The reaction mixture was incubated with NeutrAvidin agarose resin (Thermo Fisher Scientific) at 4 °C for 24 h. The agarose resin was washed three times with binding buffer. The RNA-protein complex was analyzed by using western blot with indicated antibodies. The blots in Fig. 4b–d were detected by the Licor system as follows. Blots were washed three times for 15 min with TBST and developed by using the LiCor system (LiCor CLx). Expression bands were analyzed by Image Studio Lite (LiCor 5.2.5) according to the manufacturer's manual. Note: the blot in Fig. 4a was detected by the horse radish peroxidase method as follows. Membranes were washed three times for 15 min followed by probing with secondary antibody of "peroxidase-conjugated Affinipure goat anti-mouse IgG" for Sig-1R or "peroxidase-conjugated IgG fraction monoclonal mouse anti-rabbit IgG, light-chain specific" for Nup50. Blots were washed three times for 15 min with TBST and developed by using the Azure Biosystem C600. In general, the peroxidase method offers better sensitivity over the Licor method. But the Licor method is fast with less experimental steps to perform. Depending on the need of sensitivity, some experimenters prefer Licor over the peroxidase method. But some prefer the peroxidase method as a familiar routine.

**RNA fluorescence in situ hybridization**. The RNA FISH protocol for the detection of (G4C2)$_{31}$-RNA has been reported before (Rossi et al.[5]). We performed the experiment as follows. Cells were seeded on poly-L-lysine coated coverslip and fixed with 4% paraformaldehyde in PBS before incubation at 4 °C with 70% ethanol. Cells were then rehydrated with 5 mM MgCl$_2$ in PBS and pre-hydrated with 2X SSC buffer and 10 mM sodium phosphate in 35% formamide. Commercially synthesized 250 ng/ml of Cy3-labeled (C$_4$G$_2$)$_4$ nucleotides (IDT company) were incubated with cells in 2X SSC buffer, 10 mM sodium phosphate PH 7.0, 10% dextran sulfate, 0.5 mg/ml tRNA, and 0.2% bovine serum albumin (BSA) in 35% formamide. After washing, cells were visualized by confocal microscopy (Perkin-Elmer Modular laser system 2.0 with Nikon Eclipse TE2000E microscope and Volocity version 6.3 software).

**Protein degradation assay**. Cells were cultured to 80% confluency followed by addition of cycloheximide (100–150 μg/mL, Sigma-Aldrich) to inhibit de novo protein synthesis. Cells were harvested at different time point and were lysed using the radioimmunoprecipitation assay (RIPA) lysis buffer: 10 mM Tris-HCl pH 8.0, 140 mM sodium chloride (NaCl), 0.1% sodium dodecyl sulfate (SDS), 1% Triton X-100 (Sigma-Aldrich), 1 mM ethylenediaminetetraacetic acid (EDTA), and 0.5 mM ethylene glycol tetraacetic acid (EGTA) supplemented with EDTA-free protease inhibitor cocktail (Roche). The resulting proteins were analyzed by western-blot analysis by incubating overnight at 4 °C with primary antibodies of target genes in TBST. After incubation with secondary antibodies (LiCor), blots were imaged by Odyssey infrared image system (LiCor Image Studio Lite 5.2.5). The protein turnover rate was normalized by the house-keeping gene, such as α-tubulin or actin.

**RNA isolation, reverse transcription, and quantitative real-time PCR**. The primer pairs used to perform real-time quantitative PCR are listed in Table S2 in the Supplementary Information. Total RNAs were isolated from HeLa cells by using TRIzol reagent (Invitrogen) and reverse transcribed by using Superscript III Reverse Transcriptase (Invitrogen) according to manufacturer's instructions.

Quantitative real-time PCR were performed by using SYBR Green PCR Master mix (Roche) and analyzed with ABI PRISM 7900HT sequence detection system (Applied Biosystems).

**Drosophila Stocks**. Flies were raised on standard cornmeal agar diet. The flies carrying 3 (line 370) or 30 (line 373) repeats of G4C2 hexanucleotide under the regulation of the UAS promoter were provided by Dr Peng Jin (Xu et al.[52]). Drosophila that express wild-type human Sig-1R (line Sig-1R#2) were generated by insertion of Sig-1R coding sequence between the EcoR1 and Xho1 sites of the pUAST plasmid (Couly et al.[43]). The cDNA encoding human wild-type Sig-1R was initially inserted in pCI-neo vector. After digestion by EcoR1 and XhoI restriction enzymes, the purified Sig-1R fragment was inserted between the EcoR1 and XhoI sites of the pUAST plasmid. The G304C mutation was generated by using Quickchange mutagenesis accordingly to the manufacturer's instructions (Agilent Technologies, Santa Clara, California). Germ-line -mediated P-element transformation was performed by BestGene Inc. (Chino Hills, California) in a w1118 background. The GMR-GAL4 and Elav-GAL4 (line C155) strains were obtained from the Bloomington Drosophila Stock Center (BDSC, Bloomington, Indiana) and were used to target expression specifically in the whole eye or in all neurons, respectively. Female F1 progeny that carried both UAS and GAL4 were used for subsequent analyses. In alignment with the genetic background we used the w1118 (BL5905) line from BDSC as the control. Note: all Drosophila experiments in this study were done on female flies.

**Real-time quantitative PCR: Drosophila**. Total mRNA was purified from 10 heads of flies (female, 4 days old, reared at 25 °C) in Trizol Reagent (Ambion) then submitted to trituration using plastic pestles. Chloroform (Carlo Erba) was added, and after centrifugation the upper aqueous phase was collected. RNA was then precipitated using isopropanol (VWR) and was washed in 70% ethanol and dissolved in RNase-free water. RNA concentration and purity were measured using a spectrophotometer (NanoDrop One$^c$, Thermo Scientific). RNA samples were treated with DNase from the DNA-free kit (Invitrogen) accordingly to the manufacturer's protocol. Reverse transcription was performed using M-MLV Reverse Transcriptase (Promega) following the manufacturer's instructions. Reaction plates were prepared with diluted cDNAs and Sybr No-Rox Mix (Sensifast, Bioline) by an Echo 525 acoustic liquid handler (Labcyte) and RT-qPCR experiments were performed by using a LightCycler 480 (Roche). The following primers were used for: (G4C2)30 forward 5′-GGGATCTAGCCACCATGGAG-3′ and reverse 5′-TACCGTCGACTGCAGAGATTC-3′; actin (house-keeping control gene) forward 5′- GCGCGGTTACTCTTTCACCA-3′ and reverse 5′- ATGTCACGGAC-GATTTCACG-3′. The primers for (G4C2)30 were designed to amplify a 3′ region immediately after the G4C2 repeats as previously published (Zhang et al.[12]). RT-qPCRs were conducted for 45 cycles (10 s at 95 °C, 10 s at 60 °C, and 10 s at 72 °C). Fold changes of gene expression were analyzed using the 2-ΔΔCp method. Data collected from at least four independent experiments were averaged and presented as mean ± SEM. Statistical analysis was performed using Student's $t$ test.

**Western blot: Drosophila**. Heads of flies ($n = 4$; female; four days old, reared at 25 °C) were homogenized in 50 μl RIPA lysis buffer (50 mm Tris-HCl, pH 8, 150 mm NaCl, 0.5% sodium deoxycholate, 0.1% sodium dodecyl sulfate, and 1% Igepal CA-630) supplemented with cOmpleteTM protease inhibitor cocktail (Merck, Darmstadt, Germany). Following a 1-min centrifugation, 1/4 (v/v) sample Laemmli buffer was added to supernatants. Total proteins were separated through a 10% polyacrylamide resolving gel, transferred to a nitrocellulose membrane (AmershamTM, Merck). Membrane was blocked for 1 h in the blocking solution (1X PBS, 0.1% Tween 20, and 5% dry milk) and incubated overnight with primary antibodies at 4 °C. Sig-1R protein was detected using a rabbit polyclonal antibody (1:100) generated by Abliance (Compiègne, France) and raised against residues 142-161 (KSEVFYPGETVVHGPGEATAV) of human Sig-1R. Rat anti-Elav antibody (1/700, 7E8A10, Developmental Studies Hybridoma Bank, Iowa City, Iowa) was used as a loading control. Secondary peroxidase-conjugated antibodies (1/5000, Jackson ImmunoResearch, Cambridge, UK) were incubated for 2 h in blocking solution. Chemiluminescence was revealed by using the ClarityTM Western ECL Blotting substrates (Bio-Rad) and the ChemiDoc2 Touch Imaging System (Bio-Rad).

**External eye morphology: Drosophila**. For examination of external eye phenotype, flies were reared at 29 °C. At least 48 flies from four independent groups were examined. Data are shown as the percent of flies with necrotic spots on their eyes at 15–20 days of age. Statistical analysis was performed by using the Student's $t$ test.

**Negative geotaxis test: Drosophila**. Startle-induced climbing response was assessed by using the negative geotaxis test. Flies were reared during 4 days posteclosion at 25 °C. Then they were anesthetized with CO2 and eight flies were placed in a plastic column (1.3 cm diameter × 30 cm). After 20 min recovery, columns were disposed vertically and flies were tapped to the bottom of the column. Flies that remained at the bottom or climbed above the 22 cm mark were counted after 1 min. The test was repeated three times for each batch of flies at 1 min intervals. The data are the mean of at least four trials and are presented as

percentages of flies to the top or at the bottom. Statistical significance was assessed by ANOVA followed by Tukey's multiple-comparison test.

**Electrophysiological recordings after electroconvulsive stimulation: Drosophila.** In previous studies, the bang-sensitive phenotype was associated to long firing discharge at the neuromuscular junction after a high-frequency electroconvulsive stimulation of the giant fiber pathway (Kuebler and Tanouye[56]; Lee and Wu[57]). Flies were reared 1–2 days posteclosion at 25 °C. Briefly, head and thorax of each fly were glued on a needle under $CO2$ anesthesia. Bipolar tungsten electrodes were introduced into the head to stimulate the giant fiber circuit. As a reference, an Ag/AgCl electrode was placed into the abdomen. To record evoked responses, a borosilicate glass micropipette filled with 3 M KCl was inserted into a muscular fiber of the dorsal longitudinal indirect flight muscles. Stimulation was induced by a Grass S88 Stimulator (GRASS Instruments). Recordings were made with an Intracellular Electrometer IE-210 amplifier (Warner Instruments) connected to a PowerLab 4/35. Recordings were digitized at a frequency of 20 KHz. High-frequency stimulation at 200 Hz was delivered to the brain neurons during 2 s at 30 V. Quantitative analysis was performed only on flies showing electroconvulsion by using LabChart 8 software. Data from 20 flies per condition were averaged and presented as mean ± SEM. Statistical analysis was performed using the Student's $t$ test.

**Statistics and reproducibility.** For all experiments subjected to statistical analyses, data were collected from at least three independent experiments and were compared for statistical significance by using Prism (version 8.2 at the NIDA USA lab, or version 5.01 at the INSERM France lab; GraphPad, San Diego, CA, USA). No samples were pre-allocated to specific groups to maintain randomization. Data were collected from experiments performed in replicates and were expressed as means ± standard error of means (SEM). Comparisons among multiple groups were performed for most of the experiments in this study by using a two-way ANOVA with appropriate post hoc tests. A p-value of $p \leq 0.05$ was considered statistically significant. For comparisons between non-linear regression curves (i.e., Fig. 3d–f), the second order polynomial (quadratic) model was first used for the curve fittings. Next, the 'extra sum-of-squares F-test' was used to test if the best-fit curve of a group is the same as the global (shared) fitting curve. A p-value ≤ 0.05 was considered statistically significant. Comparisons between two experimental conditions were performed by using the unpaired Student $t$ test (i.e., Fig. 8f, h, i). A p-value ≤ 0.05 was considered statistically significant. For comparisons among multiple groups with Drosophila (i.e., Fig. 8c–e, k), statistical significance was assessed by ANOVA followed by Tukey's multiple-comparison test. A p-value ≤ 0.05 was considered statistically significant.

Note: because of the limit of words in a figure legend, the statistical details of Fig. 8c–k are given as follows for the sake of clarity. (Fig. 8c) The quantification of flies presenting necrotic spots in the eyes were from four independent studies for each group with the total number of 91 in control, 48 in $(G4C2)_{30}$, and 68 in $(G4C2)_{30}$ + Sig-1R group. Data are presented as means ± SEM; $n = 4$ independent studies for each group; one-way ANOVA followed by Tukey's multiple comparisons test, 95% CI of difference for control vs. $(G4C2)_{30}$, control vs. $(G4C2)_{30}$ + Sig-1R and $(G4C2)_{30}$ vs. $(G4C2)_{30}$ + Sig-1R are −54.46 to −22.74, −18.55 to 13.18, and 20.05 to 51.78, respectively; ***$p < 0.001$. (Fig. 8d) Climbing performances of 4-day-old flies expressing no transgene (control), 3 ($(G4C2)_3$) or 30 G4C2 repeats ($(G4C2)_{30}$). Transgenes were expressed in neurons. In each experiment, the proportions of flies that climbed to the top of the column or that remained at the bottom were determined after 1 min. $n = 8$ flies per group; number of trials: control, 4; $(G4C2)_3$, 5; $(G4C2)_{30}$, 5. ***$p < 0.001$ versus control. Data are presented as means ± SEM; $n = 4–5$ trials for each group; one-way ANOVA followed by Tukey's multiple comparisons test. In the flies climbing to the top groups, 95% CI of difference for control vs. $(G4C2)_3$, control vs. $(G4C2)_{30}$ and $(G4C2)_3$ vs. $(G4C2)_{30}$ are −0.9948 to 18.08, 83.17–102.2, and 75.18–93.16, respectively. In the flies remaining at the bottom groups, 95% CI of difference for control vs. $(G4C2)_3$, control vs. $(G4C2)_{30}$ and $(G4C2)_3$ vs. $(G4C2)_{30}$ are −11.41 to 9.328, −98.91 to −78.17, and −97.28 to −77.72, respectively; ***$p < 0.001$ vs. control. (Fig. 8e) Climbing performances of 4-day-old flies expressing $(G4C2)_{30}$ alone or with Sig-1R or the green fluorescent protein GFP (GFP65T) in neurons. $n = 8$ flies per group; number of trials: $(G4C2)_{30}$, 7; $(G4C2)_{30}$ + Sig-1R, 7; $(G4C2)_{30}$ + GFP65T, 5. Data are presented as means ± SEM; one-way ANOVA followed by Tukey's multiple comparisons test. In the flies climbing to the top groups, 95% CI of difference for $(G4C2)_{30}$ vs. $(G4C2)_{30}$ + Sig-1R, $(G4C2)_{30}$ vs. $(G4C2)_{30}$ + GFP65T and $(G4C2)_{30}$ + Sig-1R vs. $(G4C2)_{30}$ + GFP65T are −30.64 to −8.642, −14.55 to 9.550, and 5.093 to 29.19, respectively. In the flies remaining at the bottom groups, 95% CI of difference for $(G4C2)_{30}$ vs. $(G4C2)_{30}$ + Sig-1R, $(G4C2)_{30}$ vs. $(G4C2)_{30}$ + GFP65T, and $(G4C2)_{30}$ + Sig-1R vs. $(G4C2)_{30}$ + GFP65T are 25.67 to 64.80, −3.458 to 39.41, and −48.70 to −5.828, respectively; ***$p < 0.001$ vs. $(G4C2)_{30}$. (Fig. 8f) Sig-1R does not modify $G4C2_{30}$ mRNA expression. Total RNAs were extracted from heads of flies expressing $G4C2_{30}$ alone or together with human Sig-1R. Transgenes were expressed in neurons. The quantitative real-time PCR was performed to measure mRNA levels of $G4C2_{30}$ by using specific primers for amplifications. The mRNA expression levels of $G4C2_{30}$ ($n = 5$) or $(G4C2)_{30}$ + Sig-1R ($n = 4$) were normalized to house-keeping gene actin. Data are presented as means ± S.E.M. Statistical analysis was performed using unpaired two-tailed $t$-test ($p = 0.2950$). (Fig. 8g) Representative traces of evoked responses after an

electroconvulsive stimulation (30 V, 200 Hz) in flies expressing no transgene (control) or $(G4C2)_{30}$. (Fig. 8h) Number of spikes in firing discharges induced by an electroconvulsive stimulation of flies expressing $(G4C2)_{30}$ alone or together with Sig-1R. Data from 20 flies were averaged and are presented as means ± S.E.M. Statistical analysis was performed using unpaired two-tailed $t$-test (*$p = 0.0385$). (Fig. 8i) Duration of firing discharges for flies expressing $(G4C2)_{30}$ alone or together with Sig-1R. Data from 20 flies were averaged and are presented as means ± S.E.M. Statistical analysis was performed using unpaired two-tailed $t$-test (*$p = 0.0375$). (Fig. 8j) Expression of Sig-1R-E102Q in retina leads to rough eye phenotype. Representative external eye morphology of 1-day aged flies expressing $G4C2_{30}$ or human Sig-1R-$E102Q$ alone or together. Note: transgenes were expressed in eyes only. (Fig. 8k) Sig-1R-$E102Q$ failed to ameliorate climbing performances of flies expressing expanded G4C2 repeats. Climbing performances of 4-day-old flies expressing no transgene (control), $G4C2_{30}$ alone or together with Sig-1R-$E102Q$ or Sig-1R-$E102Q$ alone. Note: transgenes were expressed in neurons. $n = 8$ flies per group; number of trials: Control, 6; $(G4C2)_{30}$, 7; $(G4C2)_{30}$ + Sig-1R-$E102Q$, 8; Sig-1R- $E102Q$, 6. Data are presented as means ± S.E.M. Statistical analysis was performed using ANOVA followed by Tukey's multiple-comparison test (***$p < 0.001$ versus control, ns: not significant). In the flies climbing to the top groups, 95% CI of difference for control vs $(G4C2)_{30}$, control vs $(G4C2)_{30}$ + Sig-1R-E102Q, control vs Sig-1R-E102Q, $(G4C2)_{30}$ vs $(G4C2)_{30}$ + Sig-1R-E102Q, $(G4C2)_{30}$ vs Sig-1R-E102Q, and $(G4C2)_{30}$ + Sig-1R-E102Q vs Sig-1R-E102Q are 88.03 to 101.6, 86.45 to 99.66, −4.981 to 9.148, −8.119 to 4.547, −99.57 to −85.95 and −97.58 to −84.36, respectively. In the flies remaining at the bottom groups, 95% CI of difference for control vs $(G4C2)_{30}$, control vs $(G4C2)_{30}$ + Sig-1R-E102Q, control vs Sig-1R-E102Q, $(G4C2)_{30}$ vs $(G4C2)_{30}$ + Sig-1R-E102Q, $(G4C2)_{30}$ vs Sig-1R-E102Q, and $(G4C2)_{30}$ + Sig-1R-E102Q vs Sig-1R-E102Q are −78.81 to −33.89, −67.29 to −23.68, −26.09 to 20.53, −10.03 to 31.76, 31.11 to 76.03, and 20.91 to 64.51, respectively.

**Reporting summary.** Further information on research design is available in the Nature Research Reporting Summary linked to this article.

## Data availability

The data that support this study are available from the corresponding authors upon reasonable request.

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

## Acknowledgements
We thank the very generous gifts of (G4C2)-RNA constructs from Mauro Cozzolino (Rome, Italy) and supply of NSC-34 cells from Yijuang Chern (Taipei, Taiwan). We thank Jeffrey Rothstein and Lindsey Hayes of Johns Hopkins University for helpful discussions. We thank Dr. Peng Jin (Emory University) for the UAS-(G4C2)$_3$ and UAS-(G4C2)$_{30}$ transgenic flies. We also thank Philippe Clair and Marie-Pierre Blanchard for technical advices. This study was supported by the Intramural Research Program of the National Institute on Drug Abuse of NIH/DHHS, MOST grants (108-2321-B-038-008 and 109-2636-B-038-002) of Taiwan, and Supports from University of Montpellier and INSERM of France.

## Author contributions
Conceptualization by P.T.L. and T.P.S; methodology by P.T.L., J-C.L., S.M.W., J-Y. C., W-C. C., H.E.W., T.M., and T.P.S.; investigation by P.T.L., J-C.L., S.M.W., J-Y.C., B.K., T.M., and T.P.S.; Writing - original draft by P.T.L., J-C. L., H.E.W., and T.P.S; Writing – review & editing by T.M. and T.P.S; Supervision by W-C.C., T.M., and T.P.S.

## Competing interests
The authors declare no competing interests.
