## [Peer Review File · Nature Communications]

Reviewers' comments:

Reviewer #1 (Remarks to the Author):

The paper by Lee and colleagues presents data relevant to the role of Sig-1R in nucleocytoplasmic transport and ALS. The topic is important given the fundamental biology associated with protein transport between nucleus and cytoplasm and the gravity of ALS as a neurodegenerative disease. The authors are leaders in the field of sigma 1 receptor biology and have a number of tools to interrogate function of this protein.

The paper makes a number of claims including that Sig-1R colocalizes with RAN-GAP and co-immunoprecipitates with nuclear pore proteins and that Sig1R- counteracts the nuclear-cytoplasmic deficit that underlies 40% of ALS cases.

The study is performed in HeLa cells (and in NG-108 cells), but conclusions are drawn about ALS. The work would be strengthened by inclusion of studies utilizing the C9ORF72 mouse model of ALS. Several of the models exhibit the pathophysiology and the motor phenotype. Including these in vivo studies would enhance the impact of study tremendously. Indeed using an ALS model with/without Sig1-R would allow the investigators to test their hypothesis (stated in the abstract) that Sig-1Rs may benefit ALS patients by chaperoning the nuclear pore assembly and "sponge" away deleterious G4C2-RNA repeats. Presenting in vivo data is critical if Sig-1R is truly implicated in the pathogenesis of ALS, which is the basis of the paper.

There are some specific additional points that should be considered as this work moves forward.

First, please identify the portions of the manuscript (introduction, results, discussion).

Introduction. The introduction is somewhat challenging to read. As Nature Communications appeals to a broad audience, this is particularly important. Please expand upon the information about Ran (e.g. Ran is a small Ras-related GTPase that mediates the nucleocytoplasmic exchange of macromolecules across the nuclear envelope...), as well as the nuclear proteins under study and precisely how this all relates to ALS. There is an excellent review by Ferreira that discusses nucleocytoplasmic transport and ALS, and also shows images of Ran co-localization that are performed in tissue.

Results.

Fig. 1a – in the top panel there are 3 RanGAP-positive cells shown, why does only one express Sig-1R? It would be excellent to complement the work with demonstration that Sig-1R colocalizes with this protein in the tissues that are relevant to ALS, focusing on motoneurons perhaps.

Fig. 1C – why is the Sig-1R band so weak in this immunoblot? There seem to be many antibodies used in the study, but it is not clear why several detection methods are needed (unless for confirmation/validation). Can this be clarified? Mention is made in the figure legend (fig. 1) about a "custom-made anti Sig-1R antiserum (#5460) see Methods" – however there is no information about this in the methods section.

Figure 2 – please clarify how the protein degradation assay is performed.

Figure 3 – using the HeLa cells (and the NG-108 cells) it appears that the G3C2/Sig-1R complex is not well detected, so the authors now use rat liver microsomes. What conclusion is to be drawn from data in the HeLa cells if the rat microsomes were required for this pull-down experiment? Can this be explained?

Figure 4 – panel C, is there any way to quantify the level of Ran in the cytoplasm to support the statement that Sig-R knockdown exacerbates the increase of Ran in the cytoplasm?

Figure 4 – on p. 9 there is a reference to Fig. 4E stating that a single amino acid mutation at 102 of Sig-1R reduced the ability to bind G4C2-RNA. However, this data was omitted from the figure.

Discussion is quite limited (three short paragraphs). It seems that the findings need to be placed within the larger context of the field of ALS and Sig-1R.

Reviewer #2 (Remarks to the Author):

Several years ago, nucleocytoplasmic transport was identified to play a critical role in the pathogenesis of C9-ALS/FTD, a form of ALS/FTD caused by G4C2 repeat expansion in the gene C9orf72. It was shown that G4C2 repeat mRNA interacts and sequesters RanGAP1, a critical regulator of nucleocytoplasmic transport, leading to mislocalized Ran and several nucleoporins (Nups). Ever since this discovery, defects in nucleocytoplasmic transport, such as Ran mislocalization and loss of function of Nups, have been implicated in several other forms of ALS/FTD, as well as other neurodegenerative diseases. However, it is unclear how nucleocytoplasmic transport is regulated/maintained in cells and disrupted in diseases.

Using immunofluorescent staining and biochemical approaches, Lee et al. has identified Sigma-1 Receptor (Sig-1R) as a chaperone that maintains proper nucleocytoplasmic transport in normal cells and may restore nucleocytoplasmic transport in cells expressing G4C2 repeat expansion. The authors first showed that Sig-1R localizes to the nuclear rim and interacts with RanGAP, Ran, and several Nups. Next, they showed that loss of Sig-1R by RNAi causes loss of Nups due to increased instability. Furthermore, they showed that Sig-1R biophysically and genetically interacts with G4C2 RNA. The topic of this paper is important, and the findings are interesting. But the studies are preliminary—there is no functional readout of nucleocytoplasmic transport or neurodegeneration. In addition, the findings have not been verified in an in vivo model. Furthermore, the quality of some data needs improvement.

1. C9orf72 has been implicated in both ALS and FTD. The authors should not only mention ALS, unless the studies are ALS-specific, i.e. motor-neuron-related.
2. The authors must also perform their studies in neurons and in an in vivo model system, such as mouse or fly models.
3. While loss of Nups or Ran mislocalization can disrupt nucleocytoplasmic transport, the authors need a functional readout to show that nucleocytoplasmic transport is indeed disrupted. To relate their discoveries to ALS/FTD, the authors must also include a functional readout of neuronal death/degeneration (e.g. does overexpressing Sig-1R suppresses neurodegeneration in C9 cultured neurons and/or animal models).
4. The authors argue that Sig-1R sponges G4C2, based on 1) Sig-1R interacts with G4C2 and 2) Sig-1R overexpression suppresses G4C2-mediated defects in Ran. However, this argument is not sound. This is because loss of Sig-1R disrupts Ran and Nups, suggesting a physiological role of Sig-1R in nucleocytoplasmic transport. Thus, 1) and 2) can be explained as that G4C2 disrupts nucleocytoplasmic transport via binding to and sequestering Sig-1R and that overexpressing Sig-1R suppresses G4C2-mediated Ran defects simply by restoring the physiological functions of Sig-1R, rather than absorbing the G4C2 toxicity to RanGAP, etc. It will be interesting to know what happens to Sig-1R in cells expressing G4C2, as well as in C9 patient and animal models.
5. Fig. 1b: the RanGAP/Sig-1R interaction is uninterpretable. The authors need to repeat it. The same thing applies to the Ran/Sig-1R interaction in Fig. 1c.
6. The blots in Fig.3 are uninterpretable because the input and pull-down have different experimental conditions. In Fig. 3c, the arrow indicates G4C2/Sig-1R complex, but there is no G4C2 in the input.
7. Fig. 4 needs quantification, and Fig. 4a and b also need scale bars.
8. The tubulin blots in Fig. 5a and S1 have a unique color/contrast. The authors need to improve the quality of these blots.

Responses to Reviewers' Comments (responses follow each paragraph of comments are highlighted in blue)

Reviewer #1 (Remarks to the Author):

The paper by Lee and colleagues presents data relevant to the role of Sig-1R in nucleocytoplasmic transport and ALS. The topic is important given the fundamental biology associated with protein transport between nucleus and cytoplasm and the gravity of ALS as a neurodegenerative disease. The authors are leaders in the field of sigma 1 receptor biology and have a number of tools to interrogate function of this protein.

Response: We thank the reviewer for the positive opinion on the importance of the topic of this study.

The paper makes a number of claims including that Sig-1R colocalizes with RAN-GAP and co-immunoprecipitates with nuclear pore proteins and that Sig1R- counteracts the nuclear-cytoplasmic deficit that underlies 40% of ALS cases.

The study is performed in HeLa cells (and in NG-108 cells), but conclusions are drawn about ALS. The work would be strengthened by inclusion of studies utilizing the C9ORF72 mouse model of ALS. Several of the models exhibit the pathophysiology and the motor phenotype. Including these in vivo studies would enhance the impact of study tremendously. Indeed using an ALS model with/without Sig1-R would allow the investigators to test their hypothesis (stated in the abstract) that Sig-1Rs may benefit ALS patients by chaperoning the nuclear pore assembly and “sponge” away deleterious G4C2-RNA repeats. Presenting in vivo data is critical if Sig-1R is truly implicated in the pathogenesis of ALS, which is the basis of the paper.

Response: We agree with the suggestion that in vivo data must be included to enhance the weight of this manuscript. Accordingly, we have chosen to use drosophila as an in vivo animal model for the ALS and to test out the functionality of our biochemical and cellular biological findings. The results are positive. We appreciate it very much this important suggestion by the reviewer and hope that the reviewer may agree with our efforts in this regard. Please note: we have also included NSC-34 motor neuron-like cells in our study to get closer in mimicking the ALS/FTD.

There are some specific additional points that should be considered as this work moves forward.

First, please identify the portions of the manuscript (introduction, results, discussion).

Response: Yes, done accordingly.

Introduction. The introduction is somewhat challenging to read. As Nature Communications appeals to a broad audience, this is particularly important. Please expand upon the information about Ran (e.g. Ran is a small Ras-related GTPase that mediates the

nucleocytoplasmic exchange of macromolecules across the nuclear envelope....), as well as the nuclear proteins under study and precisely how this all relates to ALS. There is an excellent review by Ferreira that discusses nucleocytoplasmic transport and ALS, and also shows images of Ran co-localization that are performed in tissue.

Response: Thank you for the reminder. We have thus “plain-texted” this manuscript as much as possible to meet the reading need of general public. This includes the suggestions by the reviewer of the wonderful review by Ferreira.

Results.

Fig. 1a – **(1)** in the top panel there are 3 RanGAP-positive cells shown, why does only one express Sig-1R? **(2)** It would be excellent to complement the work with demonstration that Sig-1R colocalizes with this protein in the tissues that are relevant to ALS, focusing on motoneurons perhaps.

Response: **(1)** In this type of “transient” transfection, it is known that not all cells will be transfected. This is because of the transfection efficiency cannot be at 100%. **(2)** Yes, accordingly we have chosen to employ the NSC-34 cells which is produced by motor neuron-enriched, embryonic mouse spinal cord cells, with mouse neuroblastoma and has been used often as a model cells for the ALS. We now demonstrate in Figure 2 that Sig-1Rs colocalize with RanGAP and Nup62 in those motor neuron-like cells.

Fig. 1C – **(1)** why is the Sig-1R band so weak in this immunoblot? **(2)** There seem to be many antibodies used in the study, but it is not clear why several detection methods are needed (unless for confirmation/validation). Can this be clarified? Mention is made in the figure legend (fig. 1) about a “custom-made anti Sig-1R antiserum (#5460) see Methods” – however there is no information about this in the methods section.

Response: **(1)** Because the affinity of Sig-1R with each protein may differ, the co-IP cannot be always yield strong Sig-1R signal. The original purpose of Figure 1b and 1c was to demonstrate the co-IP of Sig-1R and RanGAP and Ran. Unfortunately, the original data were apparently not good enough. Therefore, we decided to repeat the experiment that directly demonstrates the co-IP of Sig-1R with RanGAP and Ran in one single designed experiment. Results thus obtained serve the purpose and are better in quality (please see the new Fig. 1b).

(2) The reason different antibodies were used, for example to detect Sig-1R, can best be seen in the legend of Fig. 1 (d) as follows: “NuP50 antibody co-IPed with endogenous Sig-1R which in this experiment was detected by the Santa Cruz B5 anti-Sig-1R antibody (sc137075). Note: all other endogenous Sig-1Rs in western blot in this study was detected by custom-made anti-Sig-1R antiserum #5460 (see Methods). The two Sig-1R antibodies have been used interchangeably in the lab to reserve #5460 which is custom-made polyclonal and is limited in quantity (see Methods). Note: Santa Cruz B5 antiSig-1R is monoclonal, thus almost unlimited.” The reason for using either the peroxidase method or the Licor method

for detection is explained in Methods (on page 32 and page 34). Basically, In general, the peroxidase method offers better sensitivity over the Licor method. But the Licor method is fast with less experimental steps to perform. Depending on the need of sensitivity, some experimenters in the lab prefer Licor over the peroxidase method. But some prefer the peroxidase method as a familiar routine.

We hope this clarifies the reviewer's specific concern.

Figure 2 – please clarify how the protein degradation assay is performed.

Response: We are sorry that we did not make it clear in the original version. We have extended our description of the protein degradation study by using cycloheximide as follows in Fig. 3a: "Turnover of NuPs was then examined in a time-lapsed manner in cycloheximide-treated cells in which cycloheximide was used to stop the *de novo* synthesis of proteins. Cycloheximide is known to interfere with the translation step in protein synthesis, thus blocking translational elongation. In the presence of cycloheximide, all proteins detected by western blots are existing proteins waiting to be degraded without the presence of newly *de novo* synthesized proteins. Thus, western blots typically show a time-dependent decrease of that protein of interest. Western blotting indeed indicates a time-dependent decrease of NuP358 and NuP214 between 4-8 hours after cycloheximide (100 µg/ml) treatment (Fig. 3c)."

We hope this clarifies the method as such.

Figure 3 – using the HeLa cells (and the NG-108 cells) it appears that the G3C2/Sig-1R complex is not well detected, so the authors now use rat liver microsomes. What conclusion is to be drawn from data in the HeLa cells if the rat microsomes were required for this pull-down experiment? Can this be explained?

Response: Yes. Let us explain as follows. What we tried to demonstrate was that since the detection signal, albeit being positive, was not strong when using homogenates from whole HeLa or NG108 cells as the source of Sig-1Rs, we might as well use the liver microsome to show a stronger signal. The reason for using liver is that it contains highest level of Sig-1R among the organs examined (Hayashi and Su, Figure 4D, Cell, 2007). The reason for using microsomes is that they contain highest level of Sig-1Rs among the cellular components (Hayashi and Su, Figure 1G (note: the MAM and P3 are in the same preparation in Fig. 4b of this study), Cell, 2007).

Figure 4 – panel C, is there any way to quantify the level of Ran in the cytoplasm to support the statement that Sig-R knockdown exacerbates the increase of Ran in the cytoplasm?

Response: Yes, the "NIH Image J" system can be used to semi-quantify and compare images. We have thus used this system and quantified the images from this experiment. The results are now shown in Figure 6 of the revised manuscript.

Figure 4 – on p. 9 there is a reference to Fig. 4E stating that a single amino acid mutation at

102 of Sig-1R reduced the ability to bind G4C2-RNA. However, this data was omitted from the figure.

Response: WE are sorry for the oversight. We have now presented this data in the new Figure 4d (the third group).

Discussion is quite limited (three short paragraphs). It seems that the findings need to be placed within the larger context of the field of ALS and Sig-1R.

Response: The discussion has been extended as suggested.

Reviewer #2 (Remarks to the Author):

Several years ago, nucleocytoplasmic transport was identified to play a critical role in the pathogenesis of C9-ALS/FTD, a form of ALS/FTD caused by G4C2 repeat expansion in the gene C9orf72. It was shown that G4C2 repeat mRNA interacts and sequesters RanGAP1, a critical regulator of nucleocytoplasmic transport, leading to mislocalized Ran and several nucleoporins (Nups). Ever since this discovery, defects in nucleocytoplasmic transport, such as Ran mislocalization and loss of function of Nups, have been implicated in several other forms of ALS/FTD, as well as other neurodegenerative diseases. However, it is unclear how nucleocytoplasmic transport is regulated/maintained in cells and disrupted in diseases. Using immunofluorescent staining and biochemical approaches, Lee et al. has identified Sigma-1 Receptor (Sig-1R) as a chaperone that maintains proper nucleocytoplasmic transport in normal cells and may restore nucleocytoplasmic transport in cells expressing G4C2 repeat expansion. The authors first showed that Sig-1R localizes to the nuclear rim and interacts with RanGAP, Ran, and several Nups. Next, they showed that loss of Sig1-R by RNAi causes loss of Nups due to increased instability. Furthermore, they showed that Sig-1R biophysically and genetically interacts with G4C2 RNA.

Response: We appreciate very much on the positive notes from the reviewer.

The topic of this paper is important, and the findings are interesting. But the studies are preliminary—there is no functional readout of nucleocytoplasmic transport or neurodegeneration. In addition, the findings have not been verified in an in vivo model. Furthermore, the quality of some data needs improvement.

Response: We appreciate the reviewer's comments. We have taken new steps accordingly. Please see below.

1. C9orf72 has been implicated in both ALS and FTD. The authors should not only mention ALS, unless the studies are ALS-specific, i.e. motor-neuron-related.

Response: We thank reviewer for the suggestion. Yes, we have now mentioned FTD in the new manuscript per the reviewer's note.

2. The authors must also perform their studies in neurons and in an in vivo model system, such as mouse or fly models.

Response: Yes, accordingly we have now carried out cellular study by using NSC-34, a motor neuron-like cells, for the demonstration of Sig-1R-RanGAP/Ran colocalization (new Figure 2). Moreover, we have extended our study into in vivo animal studies by using the drosophila fly model as the reviewer kindly suggested (please see Figure 8 of the new manuscript).

3. While loss of Nups or Ran mislocalization can disrupt nucleocytoplasmic transport, the authors need a **(1) functional readout to show that nucleocytoplasmic transport is indeed disrupted**. To relate their discoveries to ALS/FTD, the authors must also include a **(2) functional readout of neuronal death/degeneration** (e.g. does overexpressing Sig-1R suppresses neurodegeneration in C9 cultured neurons and/or animal models).

Response: **(1)** In alignment with the reviewer's suggestion, we have now used the external eye morphology of drosophila as the functional readout of the (G4C2)-repeats disruption of nucleocytoplasmic transport (per Zhang et al., Nature, 2015) as seen in Figure 8a, 8b. **(2)** Per reviewer's note, we have also used drosophila as an animal model to examine the integrity of the giant fiber which controls the neuromuscular junction of the dorsal longitudinal indirect flight muscles. This model is an accepted model for ALS. We performed the negative geotaxis test (vertical climbing test; Fig. 8c, 8d), and monitored the electrophysiological response at the flight muscle (Fig. 8e, 8f, 8g). Results show that the overexpression of Sig-1Rs suppresses signs of neurodegeneration in this animal.

4. The authors argue that Sig-1R sponges G4C2, based on 1) Sig-1R interacts with G4C2 and 2) Sig-1R overexpression suppresses G4C2-mediated defects in Ran. However, this argument is not sound. This is because loss of Sig-1R disrupts Ran and Nups, suggesting a physiological role of Sig-1R in nucleocytoplasmic transport. Thus, 1) and 2) can be explained as that G4C2 disrupts nucleocytoplasmic transport via binding to and sequestering Sig-1R and that overexpressing Sig-1R suppresses G4C2-mediated Ran defects simply by restoring the physiological functions of Sig-1R, rather than absorbing the G4C2 toxicity to RanGAP, etc. It will be interesting to know what happens to Sig-1R in cells expressing G4C2, as well as in C9 patient and animal models.

Response: That is a great speculation! We thought about this possibility as well especially seeing that Sig-1Rs increase in the nucleus fraction after the (G4C2)³¹ insult (new figure Fig. 7f). But the drosophila story seen in this study offsets this speculation somewhat. The reason: drosophila has no Sig-1R of its own. May be it has a compensatory protein but no Sig-1R. Therefore, the effect of (G4C2)-repeats on drosophila cannot be on disruption of Sig-1R but may be on RanGAP whose dysfunction can be rescued by overexpression of Sig-1Rs as shown in this study. Nevertheless, we are very curious on the effect of (G4C2)-

repeats on the nuclear enrichment of Sig-1Rs. More studies are needed in the future to clarify this observation. We have also discussed this matter in the last portion of discussion.

5. Fig. 1b: the RanGAP/Sig-1R interaction is uninterpretable. The authors need to repeat it. The same thing applies to the Ran/Sig-1R interaction in Fig. 1c.

Response: The reviewer is right. We have thus repeated the experiment with a better design so now the interaction of RanGAP/Sig-1R as well as Ran/Sig-1R is clearly demonstrated in Figure 1b.

6. The blots in Fig.3 are uninterpretable because the input and pull-down have different experimental conditions. In Fig. 3c, the arrow indicates G4C2/Sig-1R complex, but there is no G4C2 in the input.

Response: A great point indeed! We have thus repeated the experiment in which the GST control is included in the experiment (new Figure 4a).

7. Fig. 4 needs quantification, and Fig. 4a and b also need scale bars.

Response: Certainly. Accordingly, Fig. 4 in the original manuscript has been replaced with Figure 5 and Figure 6 in which quantifications were performed and scale bars are now shown.

8. The tubulin blots in Fig. 5a and S1 have a unique color/contrast. The authors need to improve the quality of these blots.

Response: The quality of the blots has been improved as seen in the new Figure 7 and Figure S1.

REVIEWER COMMENTS

Reviewer #1 (Remarks to the Author):

A paper that describes the role of Sig-1R and interactions with Ran has been revised. The overall field of Sig-1R biology is important and this paper represents yet another aspect of the fascinating interactions the protein has with other proteins and offers data suggesting another key biological role for Sig-1R.

The topic is important and the study is potentially important.

In the original submission of the work, among the concerns was the absence of in vivo evidence of the relevance of the work to ALS. The authors chose to use the drosophila model to address this void. Drosophila has been used to study ALS/FTD, however as pointed out in the next-to-last paragraph of the discussion, drosophila "has no Sig-1R." While there is evidence that over-expression of Sig-1R was beneficial, it seems a curious system in which to test the overarching hypothesis of the paper. Nonetheless, if that is the system to be used, then it would strengthen the conclusion to test flies expressing the E102Q mutation (mentioned in the text). In comparison to WT, this mutant has reduced ability to bind repeat RNA, so if the hypothesis is correct over-expression this mutant version of Sig-1R will rescue neither the eye nor the climbing phenotype.

Just as a point of suggestion to strengthen the work. The newly-invited authors on the current paper (Lievens, Maurice) – who have experience with the drosophila model – have authored a paper published in Human Molecular Genetics, Volume 29, Issue 4, 15 February 2020, Pages 529–540, (<https://doi.org/10.1093/hmg/ddz267>). The quality of the work in that publication is impressive with explanations of the locomotor activity and eye development assays that tell a story and are convincing. The current paper would benefit from similar explanations of the various analyses.

The co-localization data (Fig 1) for Sig-1R and RanGAP1, Nup62 and Nup358 are intriguing. Unfortunately since the HeLa cells have to be transfected to express Sig-1R, only a few of the HeLa cells are positive for Sig-1R, while the majority are not.

It is puzzling that a cell line (that doesn't express Sig-1R) and an in vivo system (that also doesn't express Sig-1R) are the basis of a mechanistic story to enlighten as to how Sig-1R functions. There is likely a very logical explanation for this approach, but it should be stated explicitly. That is, there must be a real benefit to using these model systems that allows the biology to be interrogated.

Fig. 2 is an extensive figure, it would be helpful to fully explain the significance of the data. The individual panels (a, b, c) aren't described. Again, why are there so many more RanGAP-+ cells versus Sig-1R-+ cells?

The reference to the recent COVID-19 and Sig-1R study seems a bit of a stretch from the focus of this paper.

Reviewer #2 (Remarks to the Author):

The authors have addressed most of my concerns. Before publication, I do have two additional comments, which is critical.

1. Since fly does not have a Sig-1R gene, it is important to know that human Sig-1R, when expressed in flies, is properly folded, posttranslationally modified and/or functional. Thus, the authors need at least a Western blot detecting Sig-1R when it's overexpressed in any fly tissue or whole body. The authors should also show that the mRNA level of G4C2 repeats is not reduced when overexpressing Sig-1R to exclude the UAS/GAL4 dilution effect (see Zhang et al., 2015).

2. Since RanGAP loss of function is not the only way by which G4C2 repeats can disrupt NCT, the authors want to be more inclusive in their discussion. I suggest the authors discuss the possibility that sig1R buffers the toxicity of G4C2 repeats on other RBPs, e.g. nucleolar or stress granule proteins, as nucleolar defects can also disrupt NCT (Kwon, et al., 2014 Science and Zhang et al., 2018 Cell). In addition, the authors may also consider discussing the potential role of Sig1R in buffering DPR and/or TDP-43 toxicity, as both these toxins are shown to disrupt NCT. Especially, Sig1R has been previously linked to ALS and TDP-43.

RESPONSES TO REVIEWER COMMENTS

(Note: Authors' Responses are highlighted in blue; Figure numbers refer to those in the current revision)

Reviewer #1 (Remarks to the Author):

A paper that describes the role of Sig-1R and interactions with Ran has been revised. The overall field of Sig-1R biology is important and this paper represents yet another aspect of the fascinating interactions the protein has with other proteins and offers data suggesting another key biological role for Sig-1R.

The topic is important and the study is potentially important.

Response: We appreciate very much the positive opinion from the reviewer.

In the original submission of the work, among the concerns was the absence of in vivo evidence of the relevance of the work to ALS. The authors chose to use the *Drosophila* model to address this void. *Drosophila* has been used to study ALS/FTD, however as pointed out in the next-to-last paragraph of the discussion, *Drosophila* "has no Sig-1R." While there is evidence that over-expression of Sig-1R was beneficial, it seems a curious system in which to test the overarching hypothesis of the paper. Nonetheless, if that is the system to be used, then it would strengthen the conclusion to test flies expressing the E102Q mutation (mentioned in the text). In comparison to WT, this mutant has reduced ability to bind repeat RNA, so if the hypothesis is correct over-expression this mutant version of Sig-1R will rescue neither the eye nor the climbing phenotype.

Response: It is certainly a bit peculiar to use a system that has no endogenous Sig-1R to study the effect of Sig-1R in *Drosophila*. Nonetheless, as the reviewer mentions, the *Drosophila* has been used to study ALS/FTD and would not thus be totally irrelevant if used in the study. For example, in Couly's article cited below by the reviewer, overexpression of Sig-1R confers neuroprotection against the TPD43 toxicity. In addition, while *Drosophila* does not express endogenous Sig-1R, most of Sig-1R interactors/partners are conserved in this species.

Please note: HeLa cells do have endogenous Sig-1R as seen in Fig. 3d of the original submission. We used the HA-tagged Sig-1R for Fig. 1a simply because the signal would be stronger. Now, to make it clearer, we have performed a new experiment showing the localization of "endogenous" Sig-1Rs with RanGAP, Nup62, Nup358 in HeLa cells (Supplementary Fig. S1). Results are the same.

Regarding the effect of mutant Sig-1R on the eye. While we are keen to test the inability of the Sig-1R-E102Q mutant to rescue the deficit of eye morphology imposed by (G4C2)₃₀, we found to our surprise that the mutant by itself causes defective eyes (Fig. 8j, middle panel) and apparently worsens the defect imposed by (G4C2)₃₀ (Fig. 8j, right panel). As such, certainly no rescue was observed. The potential mechanism of the toxicity of Sig-R-E102Q per se and its relation to (G4C2)₀-repeats is also discussed (p. 18, last paragraph).

Regarding the effect of mutant Sig-1R on the climbing behavior. Per the reviewer's suggestion, we have now shown that overexpression of mutant Sig-1R-E102Q fails to rescue the climbing phenotype imposed by the (G4C2) repeats (Fig. 8k). Please see p. 14, 2nd paragraph for details.

Just as a point of suggestion to strengthen the work. The newly-invited authors on the current paper (Lievens, Maurice) – who have experience with the drosophila model – have authored a paper published in Human Molecular Genetics, Volume 29, Issue 4, 15 February 2020, Pages 529–540, (<https://doi.org/10.1093/hmg/ddz267>). The quality of the work in that publication is impressive with explanations of the locomotor activity and eye development assays that tell a story and are convincing. The current paper would benefit from similar explanations of the various analyses.

Response: We thank the reviewer's compliment on that article published by some of us. We have now incorporated explanations from that article into several parts of the Discussion of the current manuscript.

The co-localization data (Fig 1) for Sig-1R and RanGAP1, Nup62 and Nup358 are intriguing. Unfortunately since the HeLa cells have to be transfected to express Sig-1R, only a few of the HeLa cells are positive for Sig-1R, while the majority are not.

Response: Sorry for not making this clear in our previous submission. The HeLa cells in fact have endogenous Sig-1Rs (see Fig. 1d in the original manuscript). We used HA-tagged Sig-1R in the original manuscript, taking advantage of its strong signal, to examine its colocalization with endogenous RanGAP etc.

Now, to avoid potential confusion and to validate that endogenous Sig-1R colocalizes with RanGAP, Nup62, and Nup358, we now provide new image data in the supplementary showing that in HeLa cells endogenous Sig-1Rs indeed colocalize with those proteins (Supplementary Fig. S1). Please note: now Sig-1R is seen in every HeLa cell.

It is puzzling that a cell line (that doesn't express Sig-1R) and an in vivo system (that also doesn't express Sig-1R) are the basis of a mechanistic story to enlighten as to how Sig-1R functions. There is likely a very logical explanation for this approach, but it should be stated explicitly. That is, there must be a real benefit to using these model systems that allows the biology to be interrogated.

Response: Please see explanations above for the HeLa cell data. Regarding *Drosophila*, even though they do not have Sig-1Rs, their (G4C2)-repeats-induced behavioral and electrophysiological deficits can be reversed by the overexpression of Sig-1R. Those results are in alignment with results seen in HeLa cells of this study. Thus, although further studies are warranted, the *Drosophila* at present serves at least as a workable model for the present study. Please note: in support this point, Couly's article (2020) has presented extensive data showing the suitability of *Drosophila* as a model system for ALS.

Fig. 2 is an extensive figure, it would be helpful to fully explain the significance of the data. The individual panels (a, b, c) aren't described. Again, why are there so many more RanGAP++ cells versus Sig-1R++ cells?

Response: We have now fully explained in the Result the significance of the data in that the Sig-1R colocalizes with key proteins of interest related to the nuclear pore complex. We have now also explained extensively in the figure legend especially concerning individual panels (a, b, c) of the figure.

The reference to the recent COVID-19 and Sig-1R study seems a bit of a stretch from the focus of this paper.

Response: This portion of the Discussion has been taken out.

Authors-Added note to the reviewer: Figures 6c and 7b have been repeated (with results similar to the original data) because we could not locate the original PVDF cut-out segment for tubulin. Note: For blotting of multiple proteins on one PVDF membrane, we typically cut out the PVDF segment of protein of interest, per color markers as indicators, and blot each protein separately to avoid repetitive stripping of the same PVDF and to cut the waste on respective antibodies. We have also noted as such in the beginning of the Supplementary Information.

Reviewer #2 (Remarks to the Author):

The authors have addressed most of my concerns. Before publication, I do have two additional comments, which is critical.

1. Since fly does not have a Sig-1R gene, it is important to know that human Sig-1R, when expressed in flies, is properly folded, posttranslationally modified and/or functional. Thus, the authors need at least a Western blot detecting Sig-1R when it's overexpressed in any fly tissue or whole body. The authors should also show that the mRNA level of G4C2 repeats is not reduced when overexpressing Sig-1R to exclude the UAS/GAL4 dilution effect (see Zhang et al., 2015).

Response: We appreciate those important suggestions. We have now shown the western blot of overexpressed human Sig-1R in *Drosophila* (Fig. 8a).

As well, we have now shown that the mRNA of G4C2 repeats is not reduced to exclude the UAS/GAL4 dilution effect (Fig. 8f).

2. Since RanGAP loss of function is not the only way by which G4C2 repeats can disrupt NCT, the authors want to be more inclusive in their discussion. I suggest the authors discuss the possibility that sig1R buffers the toxicity of G4C2 repeats on other RBPs, e.g. nucleolar or stress granule proteins, as nucleolar defects can also disrupt NCT (Kwon, et al., 2014 Science and Zhang et al., 2018 Cell).

Response: Yes, certainly so. We have now included in Discussion (p. 17, 2nd para) the potential involvement of Sig-1R in the nucleoli concerning the stunting of ribosomal RNA biogenesis and in nucleus the mRNA maturation deficit (per Kwon et al., 2014 Science). Also, we have discussed (p. 18, 1st para) the potential role of Sig-1R in the stress granule assembly formation (per Zhang et al., 2018 Cell).

In addition, the authors may also consider discussing the potential role of Sig1R in buffering DPR and/or TDP-43 toxicity, as both these toxins are shown to disrupt NCT. Especially, Sig1R has been previously linked to ALS and TDP-43.

Response: A very interesting point indeed. We have now included in the Discussion on potential relation between Sig-1R, DPR (p. 17, last para, p. 18, 1st para), and TDP-43 (p. 18, 2nd para).

Authors-Added note to the reviewer: Figures 6c and 7b have been repeated (with results similar to the original data) because we could not locate the original PVDF cut-out segment for tubulin. Note: For blotting of multiple proteins on one PVDF membrane, we typically cut out the PVDF segment of protein of interest, per color markers as indicators, and blot each protein separately to avoid repetitive stripping of the same PVDF and to cut waste on respective antibodies. We have also noted as such in the beginning of the Supplementary Information.

REVIEWERS' COMMENTS

Reviewer #2 (Remarks to the Author):

The authors have addressed my concerns.

Response to reviewer's comment Oct 12, 2020

Reviewer #2 (Remarks to the Author):

The authors have addressed my concerns.

Response: We greatly appreciate reviewer's insightful and constructive suggestions that make this manuscript a better one to report.